# Decentralize and Randomize: Faster Algorithm for Wasserstein Barycenters

**Pavel Dvurechensky, Darina Dvinskikh**
Weierstrass Institute for Applied Analysis and Stochastics,
Institute for Information Transmission Problems RAS
{pavel.dvurechensky,darina.dvinskikh}@wias-berlin.de

**Alexander Gasnikov**
Moscow Institute of Physics and Technology,
Institute for Information Transmission Problems RAS
gasnikov@yandex.ru

**César A. Uribe**
Massachusetts Institute of Technology
cauribe@mit.edu

**Angelia Nedić**
Arizona State University,
Moscow Institute of Physics and Technology
angelia.nedich@asu.edu

## Abstract

We study the decentralized distributed computation of discrete approximations for the regularized Wasserstein barycenter of a finite set of continuous probability measures distributedly stored over a network. We assume there is a network of agents/machines/computers, and each agent holds a private continuous probability measure and seeks to compute the barycenter of all the measures in the network by getting samples from its local measure and exchanging information with its neighbors. Motivated by this problem, we develop, and analyze, a novel accelerated primal-dual stochastic gradient method for general stochastic convex optimization problems with linear equality constraints. Then, we apply this method to the decentralized distributed optimization setting to obtain a new algorithm for the distributed semi-discrete regularized Wasserstein barycenter problem. Moreover, we show explicit non-asymptotic complexity for the proposed algorithm. Finally, we show the effectiveness of our method on the distributed computation of the regularized Wasserstein barycenter of univariate Gaussian and von Mises distributions, as well as some applications to image aggregation.[1]

## 1 Introduction

Optimal transport (OT) [30, 25] has become increasingly popular in the machine learning and optimization community. Given a basis space (e.g., pixel grid) and a transportation cost function (e.g., squared Euclidean distance), the OT approach defines a distance between two objects (e.g., images), modeled as two probability measures on the basis space, as the minimal cost of transportation of the first measure to the second. Besides images, these probability measures or histograms can model other real-world objects like videos, texts, etc. The optimal transport distance leads to outstanding results in unsupervised learning [4, 7], semi-supervised learning [42], clustering [24], text classification [27], as well as in image retrieval, clustering and classification [38, 11, 39], statistics [20, 36], economics

and finance [5], condensed matter physics [8], and other applications [26]. From the computational point of view, the optimal transport distance (or Wasserstein distance) between two histograms of size $n$ requires solving a linear program, which typically requires $O(n^3 \log n)$ arithmetic operations. An alternative approach is based on entropic regularization of this linear program and application of either Sinkhorn's algorithm [11] or stochastic gradient descent [22], both requiring $O(n^2)$ arithmetic operations, which can be too costly in the large-scale context.

Given a set of objects, the optimal transport distance naturally defines their mean representative. For example, the 2-Wasserstein barycenter [2] is an object minimizing the sum of squared 2-Wasserstein distances to all objects in a set. Wasserstein barycenters capture the geometric structure of objects, such as images, better than the barycenter with respect to the Euclidean or other distances [12]. If the objects in the set are randomly sampled from some distribution, theoretical results such as central limit theorem [14] or confidence set construction [20] have been proposed, providing the basis for the practical use of Wasserstein barycenter. However, calculating the Wasserstein barycenter of $m$ measures includes repeated computation of $m$ Wasserstein distances. The entropic regularization approach was extended for this case in [6], with the proposed algorithm having a $O(mn^2)$ complexity, which can be very large if $m$ and $n$ are large. Moreover, in the large-scale setup, storage and processing of transportation plans, required to calculate Wasserstein distances, can be intractable for local computation. On the other hand, recent studies [34, 40, 37, 46, 31] on accelerated distributed convex optimization algorithms demonstrated their efficiency for convex optimization problems over arbitrary networks with inherently distributed data, i.e., the data is produced by a distributed network of sensors [35, 33, 32] or the transmission of information is limited by communication or privacy constraints, i.e., only limited amount of information can be shared across the network.

Motivated by the limited communication issue and the computational complexity of the Wasserstein barycenter problem for large amounts of data stored in a network of computers, we use the entropy regularization of the Wasserstein distance and propose a decentralized algorithm to calculate an approximation to the Wasserstein barycenter of a set of probability measures. We solve the problem in a distributed manner on a connected and undirected network of agents oblivious to the network topology. Each agent locally holds a possibly continuous probability distribution, can sample from it, and seeks to cooperatively compute the barycenter of all probability measures exchanging the information with its neighbors. We consider the semi-discrete case, which means that we fix the discrete support for the barycenter and calculate a discrete approximation for the barycenter.

**Related work.** Unlike [44], we propose a decentralized distributed algorithm for the computation of the regularized Wasserstein barycenter of a set of *continuous* measures. Working with continuous distributions requires the application of stochastic procedures like stochastic gradient method as in [22], where it is applied for regularized Wasserstein distance, but not for Wasserstein barycenter. This idea was extended to the case of non-regularized barycenter in [43, 10], where parallel algorithms were developed. The critical difference between the parallel and the decentralized setting is that, in the former, the topology of the computational network is fixed to be a star topology and it is known in advance by all the machines, forming a master/slave architecture. We seek to further scale up the barycenter computation to a huge number of input measures using *arbitrary* network topologies. Moreover, unlike [43], we use entropic regularization to take advantage of the problem smoothness and obtain faster rates of convergence for the optimization procedure. Unlike [10], we fix the support of the barycenter, which leads to a convex optimization problem and allows us to prove complexity bounds for our algorithm.

The well-developed approach based on Sinkhorn's algorithm [11, 6, 13] naturally leads to parallel algorithms. Nevertheless, its application to continuous distributions requires discretization of these distributions, leading to computational intractability when one desires good accuracy and, hence, has to use fine discretization with large $n$, which leads to the

Table 1: Summary of literature.

| PAPER | DECENTR. | CONT. | BARYC. |
|---|---|---|---|
| [11, 6, 13] | × | × | √ |
| [22] | × | √ | × |
| [43, 10] | × | √ | √ |
| OUR ALG. 2 | √ | √ | √ |

necessity of solving an optimization problem of large dimension. Thus, this approach is not directly applicable in our setting of continuous distributions, and it is not clear whether it is applicable in the decentralized distributed setting with arbitrary networks.

Recently, an alternative accelerated-gradient-based approach was shown to give better results than the Sinkhorn's algorithm for Wasserstein distance [18, 19]. Moreover, accelerated gradient methods have natural extensions for the decentralized distributed setting [40, 45, 28]. Nevertheless, existing distributed optimization algorithms can not be applied to the barycenter problem in our setting of continuous distributions as these algorithms are either designed for deterministic problems or for stochastic primal problem, whereas in our case the *dual* problem is a stochastic problem. Table 1 summarizes the existing literature on Wasserstein barycenter calculation and shows our contribution.

**Contributions.** We propose a novel algorithm for general stochastic optimization problems with linear constraints, namely the Accelerated Primal-Dual Stochastic Gradient Method (APDSGD). Based on this algorithm, we introduce a distributed algorithm for the computation of a discrete approximation for regularized Wasserstein barycenter of a set of continuous distributions stored distributedly over a network (connected and undirected) with unknown arbitrary topology. For our algorithm, we provide iteration and arithmetic operations complexity in terms of the problem parameters. Finally, we demonstrate the effectiveness of our algorithm on the distributed computation of the regularized Wasserstein barycenter of a set of von Mises distributions for various network topologies and network sizes. Moreover, we show some initial results on the problem of image aggregation for two datasets, namely, a subset of the MNIST digit dataset [29] and subset of the IXI Magnetic Resonance dataset [1].

**Paper organization.** In Section 2 we present the regularized Wasserstein barycenter problem for the semi-discrete case and its distributed computation over networks. In Section 3 we introduce a new algorithm for general stochastic optimization problems with linear constraints and analyze its convergence rate. Section 4 extends this algorithm and introduces our method for the distributed computation of regularized Wasserstein barycenter. Section 5 shows the experimental results for the proposed algorithm. The supplementary material contains the full version of the paper, including an appendix with the proofs, as well as additional results of numerical experiments.

**Notation.** We define $\mathcal{M}_+^1(\mathcal{X})$ the set of positive Radon probability measures on a metric space $\mathcal{X}$, and $S_1(n) = \{a \in \mathbb{R}_+^n \mid \sum_{l=1}^n a_l = 1\}$ the probability simplex. We denote by $\delta(x)$ the Dirac measure at point $x$, and $\otimes$ the Kronecker product. We refer to $\lambda_{\max}(W)$ as the maximum eigenvalue of a symmetric matrix $W$. We use bold symbols for stacked vectors $\mathbf{p} = [p_1^T, \cdots, p_m^T]^T \in \mathbb{R}^{mn}$, where $p_1, ..., p_m \in \mathbb{R}^n$. In this case $[\mathbf{p}]_i = p_i$ – the $i$-th block of $\mathbf{p}$. For a vector $\lambda \in \mathbb{R}^n$, we denote by $[\lambda]_l$ its $l$-th component. We refer to the Euclidean norm of a vector $\|p\|_2 := \sum_{l=1}^n ([p]_l)^2$ as 2-norm.

## 2 The Distributed Wasserstein Barycenter Problem

In this section, we present the problem of decentralized distributed computation of regularized Wasserstein barycenters for a family of possibly continuous probability measures distributed over a network. First, we provide the necessary background for regularized Wasserstein distance and barycenter. Then, we give the details of the distributed formulation of the optimization problem defining Wasserstein barycenter, which is a minimization problem with linear equality constraint. To deal with this constraint, we make a transition to the dual problem, which, as we show, due to the presence of continuous distributions, is a smooth stochastic optimization problem.

**Regularized semi-discrete formulation of optimal transport problem.** We consider entropic regularization for the optimal transport problem and the corresponding regularized Wasserstein distance and barycenter [11]. Let $\mu \in \mathcal{M}_+^1(\mathcal{Y})$ with density $q(y)$, and a discrete probability measure $\nu = \sum_{i=1}^n [p]_i \delta(z_i)$ with weights given by vector $p \in S_1(n)$ and finite support given by points $z_1, \ldots, z_n \in \mathcal{Z}$ from a metric space $\mathcal{Z}$. The regularized Wasserstein distance in semi-discrete setting between continuous measure $\mu$ and discrete measure $\nu$ is defined as[2]

$$\mathcal{W}_\gamma(\mu, \nu) = \min_{\pi \in \Pi(\mu, \nu)} \left\{ \sum_{i=1}^n \int_{\mathcal{Y}} c_i(y) \pi_i(y) dy + \gamma \sum_{i=1}^n \int_{\mathcal{Y}} \pi_i(y) \log\left(\frac{\pi_i(y)}{\xi}\right) dy \right\}, \quad (1)$$

where $c_i(y) = c(z_i, y)$ is a cost function for transportation of a unit of mass from point $z_i \in \mathcal{Z}$ to point $y \in \mathcal{Y}$, $\xi$ is the uniform distribution on $\mathcal{Y} \times \mathcal{Z}$, and the set of admissible coupling measures $\pi$ is defined as

$$\Pi(\mu, \nu) = \left\{ \pi \in \mathcal{M}_+^1(\mathcal{Y}) \times S_1(n) : \sum_{i=1}^n \pi_i(y) = q(y), y \in \mathcal{Y}, \int_{\mathcal{Y}} \pi_i(y) dy = p_i, \forall\, i = 1, \ldots, n \right\}.$$

For a set of measures $\mu_i \in \mathcal{M}_+^1(\mathcal{Z})$, $i = 1, \ldots, m$, we fix the support $z_1, \ldots, z_n \in \mathcal{Z}$ of their regularized Wasserstein barycenter $\nu$ and wish to find it in the form $\nu = \sum_{i=1}^n [p]_i \delta(z_i)$, where $p \in S_n(1)$. Then the regularized Wasserstein barycenter in the semi-discrete setting is defined as the solution to the following convex optimization problem[3]

$$\min_{p \in S_1(n)} \sum_{i=1}^m \mathcal{W}_{\gamma, \mu_i}(p) = \min_{\substack{p_1 = \cdots = p_m \\ p_1, \ldots, p_m \in S_1(n)}} \sum_{i=1}^m \mathcal{W}_{\gamma, \mu_i}(p_i), \tag{2}$$

where we used notation $\mathcal{W}_{\gamma, \mu}(p) := \mathcal{W}_\gamma(\mu, \nu)$ for fixed probability measure $\mu$.

**Network constraints in the distributed barycenter problem.** We now describe the distributed optimization setting for solving the second problem in (2). We assume that each measure $\mu_i$ is held by an agent $i$ on a network and this agent can sample from this measure. We model such a network as a fixed *connected undirected graph* $\mathcal{G} = (V, E)$, where $V$ is the set of $m$ nodes, and $E$ is the set of edges. We assume that the graph $\mathcal{G}$ does not have self-loops. The network structure imposes information constraints; specifically, each node $i$ has access to $\mu_i$ only and can exchange information only with its immediate neighbors, i.e., nodes $j$ s.t. $(i, j) \in E$.

We represent the communication constraints imposed by the network by introducing a single equality constraint instead of $p_1 = \cdots = p_m$ in (2). To do so, we define the Laplacian matrix $\bar{W} \in \mathbb{R}^{m \times m}$ of the graph $\mathcal{G}$ such that a) $[\bar{W}]_{ij} = -1$ if $(i, j) \in E$, b) $[\bar{W}]_{ij} = \deg(i)$ if $i = j$, c) $[\bar{W}]_{ij} = 0$ otherwise. Here $\deg(i)$ is the degree of the node $i$, i.e., the number of neighbors of the node. Finally, define the communication matrix (also referred to as an interaction matrix) by $W := \bar{W} \otimes I_n$.

Assuming that $\mathcal{G}$ is undirected and connected, the Laplacian matrix $\bar{W}$ is symmetric and positive semidefinite. Furthermore, the vector $\mathbf{1}$ is the unique (up to a scaling factor) eigenvector associated with the zero eigenvalue. $W$ inherits the properties of $\bar{W}$, i.e., it is symmetric and positive semidefinite. Moreover, $\sqrt{W}\mathbf{p} = 0$ if and only if $p_1 = \cdots = p_m$, where we defined stacked column vector $\mathbf{p} = [p_1^T, \cdots, p_m^T]^T \in \mathbb{R}^{mn}$. Using this fact, we equivalently rewrite problem (2) as the maximization problem with linear equality constraint

$$\max_{p_1, \ldots, p_m \in S_1(n),\ \sqrt{W}\mathbf{p}=0} - \sum_{i=1}^m \mathcal{W}_{\gamma, \mu_i}(p_i). \tag{3}$$

**Dual formulation of the barycenter problem.** Given that problem (3) is an optimization problem with linear constraints, we introduce a stacked vector of dual variables $\boldsymbol{\lambda} = [\lambda_1^T, \cdots, \lambda_m^T]^T \in \mathbb{R}^{mn}$ for the constraints $\sqrt{W}\mathbf{p} = 0$ in (3). Then, the Lagrangian dual problem for (3) is

$$\min_{\boldsymbol{\lambda} \in \mathbb{R}^{mn}} \max_{p_1, \ldots, p_m \in S_1(n)} \left\{ \sum_{i=1}^m \langle \lambda_i, [\sqrt{W}\mathbf{p}]_i \rangle - \mathcal{W}_{\gamma, \mu_i}(p_i) \right\} = \min_{\boldsymbol{\lambda} \in \mathbb{R}^{mn}} \sum_{i=1}^m \mathcal{W}^*_{\gamma, \mu_i}([\sqrt{W}\boldsymbol{\lambda}]_i), \tag{4}$$

where $[\sqrt{W}\mathbf{p}]_i$ and $[\sqrt{W}\boldsymbol{\lambda}]_i$ denote the $i$-th $n$-dimensional block of vectors $\sqrt{W}\mathbf{p}$ and $\sqrt{W}\boldsymbol{\lambda}$ respectively, the equality $\sum_{i=1}^m \langle \lambda_i, [\sqrt{W}\mathbf{p}]_i \rangle = \sum_{i=1}^m \langle [\sqrt{W}\boldsymbol{\lambda}]_i, p_i \rangle$ was used, and $\mathcal{W}^*_{\gamma, \mu_i}(\cdot)$ is the Fenchel-Legendre transform of $\mathcal{W}_{\gamma, \mu_i}(p_i)$. The following Lemma states that each $\mathcal{W}^*_{\gamma, \mu_i}(\cdot)$ is a smooth function with Lipschitz-continuous gradient and can be expressed as an expectation of a function of additional random argument.

**Lemma 1.** *Given $\mu \in \mathcal{M}_+^1(\mathcal{Y})$ with density $q(\cdot)$, the Fenchel-Legendre conjugate for $\mathcal{W}_{\gamma, \mu}(p)$ is*

$$\mathcal{W}^*_{\gamma, \mu}(\bar{\lambda}) = \mathbb{E}_{Y \sim \mu}\, \gamma \log \left( \frac{1}{q(Y)} \sum_{l=1}^n \exp \left( \frac{[\bar{\lambda}]_l - c_l(Y)}{\gamma} \right) \right),$$

*and its gradient is $1/\gamma$-Lipschitz-continuous w.r.t. 2-norm.*

Denote $\bar{\boldsymbol{\lambda}} = \sqrt{W}\boldsymbol{\lambda} = [[\sqrt{W}\boldsymbol{\lambda}]_1^T, \ldots, [\sqrt{W}\boldsymbol{\lambda}]_m^T]^T = [\bar{\lambda}_1^T, \ldots, \bar{\lambda}_m^T]^T$ and $\mathcal{W}_\gamma^*(\boldsymbol{\lambda})$ – the dual objective in the r.h.s. of (4). Then, by the chain rule, the $l$-th $n$-dimensional block of $\nabla \mathcal{W}_\gamma^*(\boldsymbol{\lambda})$ is

$$\left[\nabla \mathcal{W}_\gamma^*(\boldsymbol{\lambda})\right]_l = \left[\nabla \sum_{i=1}^m \mathcal{W}_{\gamma,\mu_i}^*([\sqrt{W}\boldsymbol{\lambda}]_i)\right]_l = \sum_{j=1}^m \sqrt{W}_{lj} \nabla \mathcal{W}_{\gamma,\mu_j}^*(\bar{\lambda}_j),\ l = 1, \ldots, m. \quad (5)$$

It follows from (5) and Lemma 1 that the *dual* problem (4) is a smooth stochastic convex optimization problem. This is in contrast to [28], where the primal problem is a stochastic optimization problem. Moreover, as opposed to the existing literature on stochastic convex optimization, we not only need to solve the dual problem but also need to reconstruct an approximate solution for the primal problem (3), which is the barycenter. In the next section, we develop a novel accelerated primal-dual stochastic gradient method for a general smooth stochastic optimization problem, which is dual to some optimization problem with linear equality constraints. Furthermore, in Section 4, we apply our general algorithm to the particular case of primal-dual pair of problems (3) and (4).

## 3  General Primal-Dual Framework for Stochastic Optimization

In this section, we consider a general smooth stochastic convex optimization problem which is dual to some optimization problem with linear equality constraints. Extending our works [16, 21, 9, 17, 19, 3, 18], we develop a novel algorithm for its solution and reconstruction of the primal variable together with convergence rate analysis. Unlike prior works, we consider the stochastic primal-dual pair of problems and one of our contributions consists in providing a primal-dual extension of the accelerated stochastic gradient method. We believe that our algorithm can be used for problems other than regularized Wasserstein barycenter problem and, thus, we, first, provide a general algorithm and, then, apply it to the barycenter problem. We introduce new notation since this section is independent of the others and is focused on a general optimization problem.

**General setup.** For any finite-dimensional real vector space $E$, we denote by $E^*$ its dual, by $\|\cdot\|$ a norm on $E$ and by $\|\cdot\|_*$ the norm on $E^*$ which is dual to $\|\cdot\|$, i.e. $\|\lambda\|_* = \max_{\|x\| \le 1}\langle \lambda, x\rangle$. For a linear operator $A : E_1 \to E_2$, the adjoint operator $A^T : E_2^* \to E_1^*$ in defined by $\langle u, Ax\rangle = \langle A^T u, x\rangle, \quad \forall u \in E_2^*, \quad x \in E_1$. We say that a function $f : E \to \mathbb{R}$ has a $L$-Lipschitz-continuous gradient w.r.t. norm $\|\cdot\|_*$ if it is continuously differentiable and its gradient satisfies Lipschitz condition $\|\nabla f(x) - \nabla f(y)\|_* \le L\|x - y\|, \quad \forall x, y \in E$.

Our main goal in this section, is to provide an algorithm for a primal-dual (up to a sign) pair of problems

$$(P) \qquad \min_{x \in Q \subseteq E} \{f(x) : Ax = b\}, \qquad (D) \quad \min_{\lambda \in \Lambda} \left\{\langle \lambda, b\rangle + \max_{x \in Q}\left(-f(x) - \langle A^T\lambda, x\rangle\right)\right\}.$$

where $Q$ is a simple closed convex set, $A : E \to H$ is given linear operator, $b \in H$ is given, $\Lambda = H^*$. We define $\varphi(\lambda) := \langle \lambda, b\rangle + \max_{x \in Q}\left(-f(x) - \langle A^T\lambda, x\rangle\right) = \langle \lambda, b\rangle + f^*(-A^T\lambda)$ and assume it to be smooth with $L$-Lipschitz-continuous gradient. Here $f^*$ is the Fenchel-Legendre dual for $f$. We also assume that $f^*(-A^T\lambda) = \mathbb{E}_\xi F^*(-A^T\lambda, \xi)$, where $\xi$ is random vector and $F^*$ is the Fenchel-Legendre conjugate function to some function $F(x, \xi)$, i.e. it satisfies $F^*(-A^T\lambda, \xi) = \max_{x \in Q}\{\langle -A^T\lambda, x\rangle - F(x, \xi)\}$. $F^*(\bar{\lambda}, \xi)$ is assumed to be smooth and, hence $\nabla_{\bar{\lambda}} F^*(\bar{\lambda}, \xi) = x(\bar{\lambda}, \xi)$, where $x(\bar{\lambda}, \xi)$ is the solution of the maximization problem $x(\bar{\lambda}, \xi) = \arg\max_{x \in Q}\{\langle \bar{\lambda}, x\rangle - F(x, \xi)\}$. Under these assumptions, the dual problem $(D)$ can be accessed by a stochastic oracle $(\Phi(\lambda, \xi), \nabla\Phi(\lambda, \xi))$ satisfying $\mathbb{E}_\xi\Phi(\lambda, \xi) = \varphi(\lambda)$, $\mathbb{E}_\xi\nabla\Phi(\lambda, \xi) = \nabla\varphi(\lambda)$, which we use in our algorithm.

**Accelerated primal-dual stochastic gradient method.** Next, we provide an accelerated algorithm for the primal-dual pair of problems $(P) - (D)$. The idea is to apply accelerated stochastic gradient method to the dual problem $(D)$, endow it with a step in the primal space and show that the new algorithm allows also approximating the solution to the primal problem. We additionally assume that the variance of the stochastic approximation $\nabla\Phi(\lambda, \xi)$ for the gradient of $\varphi$ can be controlled and made as small as we desire. This can be done, for example by mini-batching the stochastic approximation. Finally, since $\nabla\Phi(\lambda, \xi) = b - A\nabla F^*(-A^T\lambda, \xi) = b - Ax(-A^T\lambda, \xi)$, on each iteration, to find $\nabla\Phi(\lambda, \xi)$ we find the vector $x(-A^T\lambda, \xi)$ and use it for the primal iterates.

**Theorem 1.** *Let $\varphi$ have L-Lipschitz-continuous gradient w.r.t. 2-norm and $\|\lambda^*\|_2 \leq R$, where $\lambda^*$ is a solution of dual problem $(D)$. Given desired accuracy $\varepsilon$, assume that, at each iteration of Algorithm 1, the stochastic gradient $\nabla\Phi(\lambda_k, \xi_k)$ is chosen in such a way that $\mathbb{E}_\xi \|\nabla\Phi(\lambda_k, \xi_k) - \nabla\varphi(\lambda_k)\|_2^2 \leq \frac{\varepsilon L \alpha_k}{C_k}$. Then, for any $\varepsilon > 0$ and $N \geq 0$, and expectation $\mathbb{E}$ w.r.t. all the randomness $\xi_1, \ldots, \xi_N$, the output $\hat{x}_N$ generated by the Algorithm 1 satisfies*

$$f(\mathbb{E}\hat{x}_N) - f^* \leq \frac{16LR^2}{N^2} + \frac{\varepsilon}{2} \quad \text{and} \quad \|A\mathbb{E}\hat{x}_N - b\|_2 \leq \frac{16LR}{N^2} + \frac{\varepsilon}{2R}. \tag{6}$$

In step 7 of Algorithm 1 we can use a batch of size $M$ and $\frac{1}{M}\sum_{r=1}^{M} x(\lambda_{k+1}, \xi_{k+1}^r)$ to update $\hat{x}_{k+1}$. Then, under reasonable assumptions, $\hat{x}_N$ concentrates around $\mathbb{E}\hat{x}_N$ [23] and, if $f$ is Lipschitz-continuous, we obtain that (6) holds with large probability with $\hat{x}_N$ instead of $\mathbb{E}\hat{x}_N$.

## 4 Solving the Barycenter Problem

In this section, we apply the general algorithm APDSGD to solve the primal-dual pair of problems (3)-(4) and approximate the regularized Wasserstein barycenter which is a solution to (3). First, in Lemma 2, we make several technical steps to take care of the assumption of Theorem (1). Then, we introduce a change of dual variable so that the step 5 of Algorithm 1 becomes feasible for decentralized distributed setting. After that, we provide our algorithm for regularized Wasserstein barycenter problem with its complexity analysis.

---
**Algorithm 1** Accelerated Primal-Dual Stochastic Gradient Method (APDSGD)

---
**Input:** Number of iterations $N$.
1: $C_0 = \alpha_0 = 0$, $\eta_0 = \zeta_0 = \lambda_0 = \hat{x}_0 = 0$.
2: **for** $k = 0, \ldots, N-1$ **do**
3:     Find $\alpha_{k+1} > 0$ from $C_{k+1} := C_k + \alpha_{k+1} = 2L\alpha_{k+1}^2$.
       $\tau_{k+1} = \alpha_{k+1}/C_{k+1}$.
4:     $\lambda_{k+1} = \tau_{k+1}\zeta_k + (1 - \tau_{k+1})\eta_k$
5:     $\zeta_{k+1} = \zeta_k - \alpha_{k+1}\nabla\Phi(\lambda_{k+1}, \xi_{k+1})$.
6:     $\eta_{k+1} = \tau_{k+1}\zeta_{k+1} + (1 - \tau_{k+1})\eta_k$.
7:     $\hat{x}_{k+1} = \tau_{k+1}x(\lambda_{k+1}, \xi_{k+1}) + (1 - \tau_{k+1})\hat{x}_k$.
8: **end for**
**Output:** The points $\hat{x}_N, \eta_N$.

---

**Lemma 2.** *The gradient of the objective function $\mathcal{W}_\gamma^*(\boldsymbol{\lambda})$ in the dual problem* (4) *is $\lambda_{\max}(W)/\gamma$-Lipschitz-continuous w.r.t. 2-norm. If its stochastic approximation is defined as*

$$[\widetilde{\nabla}\mathcal{W}_\gamma^*(\boldsymbol{\lambda})]_i = \sum_{j=1}^m \sqrt{W}_{ij}\widetilde{\nabla}\mathcal{W}_{\gamma,\mu_j}^*(\bar{\lambda}_j), \; i = 1, ..., m, \; with$$

$$\widetilde{\nabla}\mathcal{W}_{\gamma,\mu_j}^*(\bar{\lambda}_j) = \frac{1}{M}\sum_{r=1}^M p_j(\bar{\lambda}_j, Y_r^j), \; and \; [p_j(\bar{\lambda}_j, Y_r^j)]_l = \frac{\exp(([\bar{\lambda}_j]_l - c_l(Y_r^j))/\gamma)}{\sum_{\ell=1}^n \exp(([\bar{\lambda}_j]_\ell - c_\ell(Y_r^j))/\gamma)}. \tag{7}$$

*where $M$ is the batch size, $\bar{\lambda}_j := [\sqrt{W}\boldsymbol{\lambda}]_j$, $j = 1, ..., m$, $Y_1^j, ..., Y_r^j$ is a sample from the measure $\mu_j$, $j = 1, ..., m$. Then $\mathbb{E}_{Y_r^j \sim \mu_j, j=1,...,m, r=1,...,M}\widetilde{\nabla}\mathcal{W}_\gamma^*(\boldsymbol{\lambda}) = \nabla\mathcal{W}_\gamma^*(\boldsymbol{\lambda})$ and*

$$\mathbb{E}_{Y_r^j \sim \mu_j, j=1,...,m, r=1,...,M}\|\widetilde{\nabla}\mathcal{W}_\gamma^*(\boldsymbol{\lambda}) - \nabla\mathcal{W}_\gamma^*(\boldsymbol{\lambda})\|_2^2 \leq \lambda_{\max}(W)m/M, \; \boldsymbol{\lambda} \in \mathbb{R}^{mn}. \tag{8}$$

Based on this lemma, we see that if, on each iteration of Algorithm 1, the mini-batch size $M_k$ satisfies $M_k \geq \frac{m\gamma C_k}{\alpha_k \varepsilon}$, the assumptions of Theorem 1 hold.

For the particular problem (4) the step 5 of Algorithm 1 can be written block-wise $[\boldsymbol{\zeta}_{k+1}]_i = [\boldsymbol{\zeta}_k]_i - \alpha_{k+1}\sum_{j=1}^m \sqrt{W}_{ij}\widetilde{\nabla}\mathcal{W}_{\gamma,\mu_j}^*([\sqrt{W}\boldsymbol{\lambda}_{k+1}]_j), i = 1, ..., m$. Unfortunately, this update can not be made in the decentralized setting since the sparsity pattern of $\sqrt{W}_{ij}$ can be different from $W_{ij}$ and this will require some agents to get information not only from their neighbors. To overcome this obstacle, we change the variables and denote $\bar{\boldsymbol{\lambda}} = \sqrt{W}\boldsymbol{\lambda}$, $\bar{\boldsymbol{\eta}} = \sqrt{W}\boldsymbol{\eta}$, $\bar{\boldsymbol{\zeta}} = \sqrt{W}\boldsymbol{\zeta}$. Then the step 5 of Algorithm 1 becomes $[\bar{\boldsymbol{\zeta}}_{k+1}]_i = [\bar{\boldsymbol{\zeta}}_k]_i - \alpha_{k+1}\sum_{j=1}^m W_{ij}\widetilde{\nabla}\mathcal{W}_{\gamma,\mu_j}^*([\bar{\boldsymbol{\lambda}}_{k+1}]_j), i = 1, ..., m$.

**Theorem 2.** *Under the assumptions of Section 2, Algorithm 2 after $N = \sqrt{16\lambda_{\max}(W)R^2/(\varepsilon\gamma)}$ iterations returns an approximation $\hat{p}_N$ for the barycenter, which satisfies*

$$\sum_{i=1}^m \mathcal{W}_{\gamma,\mu_i}(\mathbb{E}[\hat{p}_N]_i) - \sum_{i=1}^m \mathcal{W}_{\gamma,\mu_i}([p^*]_i) \leq \varepsilon, \quad \|\sqrt{W}\mathbb{E}\hat{p}_N\|_2 \leq \varepsilon/R. \tag{9}$$

*The total complexity is $O\left(mn \max\left\{\sqrt{\frac{\lambda_{\max}(W)R^2}{\varepsilon\gamma}}, \frac{\lambda_{\max}(W)mR^2}{\varepsilon^2}\right\}\right)$ arithmetic operations.*

We underline that even if the measures $\mu_i, i = 1, ..., m$ are discrete with large support size, it can be more efficient to apply our stochastic algorithm than a deterministic algorithm. We now explain it in more details. If a measure $\mu$ is discrete, then $\mathcal{W}^*_{\gamma,\mu}(\bar{\lambda})$ in Lemma 1 is represented as a finite expectation, i.e., a sum of functions instead of an integral, and can be found explicitly. In the same way, its gradient and, hence, $\nabla\mathcal{W}^*_\gamma(\boldsymbol{\lambda})$ in (5) can be found explicitly in a deterministic way. Then a deterministic accelerated decentralized algorithm can be applied to approximate the regularized barycenter. Let us assume for simplicity that the support of measure $\mu$ is of the size $n$. Then the calculation of the exact gradient of $\mathcal{W}^*_{\gamma,\mu}(\bar{\lambda})$ requires $O(n^2)$ arithmetic operations and the overall complexity of the deterministic algorithm is $O\left(mn^2\sqrt{\lambda_{\max}(W)R^2/\gamma\varepsilon}\right)$. For

**Algorithm 2** Distributed computation of Wasserstein barycenter

**Input:** Each agent $i \in V$ is assigned its measure $\mu_i$.
1: All agents set $[\bar{\boldsymbol{\eta}}_0]_i = [\bar{\boldsymbol{\zeta}}_0]_i = [\bar{\boldsymbol{\lambda}}_0]_i = [\hat{\mathrm{p}}_0]_i = \mathbf{0} \in \mathbb{R}^n$, $C_0 = \alpha_0 = 0$ and $N$
2: For each agent $i \in V$:
3: **for** $k = 0, \ldots, N-1$ **do**
4:     Find $\alpha_{k+1} > 0$ from $C_{k+1} := C_k + \alpha_{k+1} = 2L\alpha_{k+1}^2$. $\tau_{k+1} = \alpha_{k+1}/C_{k+1}$.
5:     Set $M_{k+1} = \max\{1, \lceil m\gamma C_{k+1}/(\alpha_{k+1}\varepsilon)\rceil\}$
6:     $[\bar{\boldsymbol{\lambda}}_{k+1}]_i = \tau_{k+1}[\bar{\boldsymbol{\zeta}}_k]_i + (1-\tau_{k+1})[\bar{\boldsymbol{\eta}}_k]_i$
7:     Generate $M_{k+1}$ samples $\{Y_r^i\}_{r=1}^{M_{k+1}}$ from the measure $\mu_i$ and set $\widetilde{\nabla}\mathcal{W}^*_{\gamma,\mu_i}([\bar{\boldsymbol{\lambda}}_{k+1}]_i)$ as in (7).
8:     Share $\widetilde{\nabla}\mathcal{W}^*_{\gamma,\mu_i}([\bar{\boldsymbol{\lambda}}_{k+1}]_i)$ with $\{j \mid (i,j) \in E\}$
9:     $[\bar{\boldsymbol{\zeta}}_{k+1}]_i = [\bar{\boldsymbol{\zeta}}_k]_i - \alpha_{k+1}\sum_{j=1}^m W_{ij}\widetilde{\nabla}\mathcal{W}^*_{\gamma,\mu_j}([\bar{\boldsymbol{\lambda}}_{k+1}]_j)$
10:     $[\bar{\boldsymbol{\eta}}_{k+1}]_i = \tau_{k+1}[\bar{\boldsymbol{\zeta}}_{k+1}]_i + (1-\tau_{k+1})[\bar{\boldsymbol{\eta}}_{k+1}]_i$
11:     $[\hat{\mathrm{p}}_{k+1}]_i = \tau_{k+1}p_i([\bar{\boldsymbol{\lambda}}_{k+1}]_i, Y_1^i) + (1-\tau_{k+1})[\hat{\mathrm{p}}_{k+1}]_i$, where $p_i(\cdot, \cdot)$ is defined in (7).[4]
12: **end for**
**Output:** $\hat{\mathrm{p}}_N$.

comparison, the complexity of our randomized approach in Theorem 2 is proportional to $n$, but not to $n^2$. So, our randomized approach is superior in the regime of large $n$.

## 5 Experimental Results

In this section, we present experimental results for Algorithm 2. Initially, we consider a set of agents over a network, where each agent $i$ can samples from a privately held random variable $Y_i \sim \mathcal{N}(\theta_i, v_i^2)$, where $\mathcal{N}(\theta, v^2)$ is a univariate Gaussian distribution with mean $\theta$ and variance $v^2$. Moreover, we set $\theta_i \in [-4, 4]$ and $v_i \in [0.1, 0.6]$. The objective is to compute a discrete distribution $p \in S_1(n)$ that solves (2). We assume $n = 100$ and the support of $p$ is a set of 100 equally spaced points on the segment $[-5, 5]$. Figure 1 shows the performance of Algorithm 2 for four classes of networks: complete, cycle, star, and Erdős-Rényi. Moreover, we show the behavior for different network sizes, namely: $m = 10, 100, 200, 500$. Particularly we use two metrics: the function value of the dual problem and the distance to consensus, i.e., $\mathcal{W}^*_\gamma(\boldsymbol{\lambda})$ and $C(\hat{\mathrm{p}}) := \|\sqrt{W}\hat{\mathrm{p}}\|_2$. As expected, when the network is a complete graph, the convergence to the final value and the distance to consensus decreases rapidly. Nevertheless, the performance in graphs with degree regularity, such as the cycle graph and the Erdős-Rényi random graph, is similar to a complete graph with much less communication overhead. For the star graph, which has the worst case between the maximum and minimum number of neighbors among all nodes, the algorithm performs poorly. Figure 2 shows the convergence of the local barycenter of a set of von Mises distributions. Each agent over an Erdős-Rényi random graph can access private realizations from a von Mises random variable. Particularly, for the cases of von Mises distributions, we have used the angle between two points distance function. Figure 3 shows the computed local barycenter of 9 agents in a network of 500 nodes at different iteration numbers. Each agent holds a local copy of a sample of the digit 2 ($56 \times 56$ image) from the MNIST dataset [29]. All agents converge to the same image that structurally represents the aggregation of the original 500 images held over the network. Finally, Figure 4 shows a simple example of an application of Wasserstein barycenter on medical image aggregation where we have 4 agents connected over a cycle graph and each agent holds a magnetic resonance image ($256 \times 256$) from the IXI dataset [1].

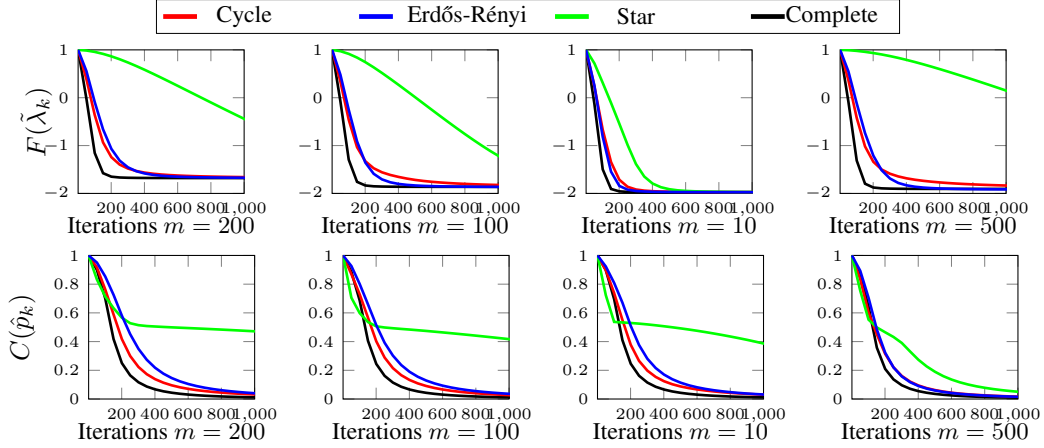

Figure 1: Dual function value and distance to consensus for $200, 100, 10, 500$ agents, $M_k = 100$ and $\gamma = 0.1$.

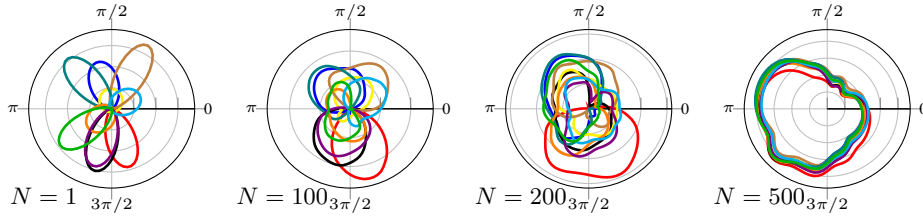

Figure 2: Wasserstein barycenter of von Mises distributions for 10 agents at different iteration numbers.

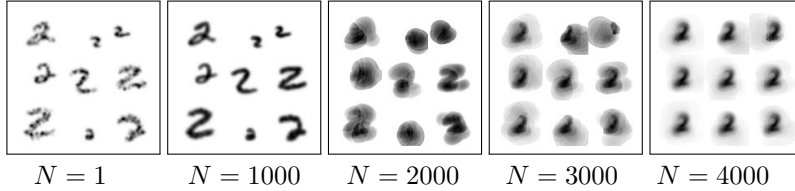

Figure 3: Wasserstein barycenter of digit 2 from the MNIST dataset [29]. Each block shows a subset of 9 randomly selected local barycenters at different time instances.

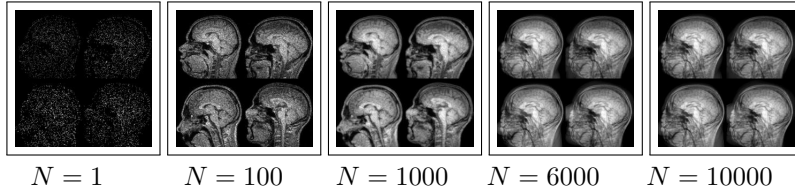

Figure 4: Wasserstein barycenter for a subset of images from the IXI dataset [1]. Each block shows the local barycenters of 4 agents at different time instances.

## 6  Conclusions

We propose a novel distributed algorithm for regularized Wasserstein barycenter problem for a set of continuous measures stored distributedly over a network of agents. Our algorithm is based on a new general algorithm for the solution of stochastic convex optimization problems with linear constraints. In contrast to the recent literature, our algorithm can be executed over arbitrary connected and static networks where nodes are oblivious to the network topology, which makes it suitable for large-scale network optimization setting. Additionally, our analysis indicates that the randomization strategy provides faster convergence rates than the deterministic procedure when the support size of the barycenter is large. The implementation of our algorithm on real networks, requires further work, as well as its extension to the decentralized distributed setting of Sinkhorn-type algorithms [6] for regularized Wasserstein barycenter and other related algorithms, e.g., Wasserstein propagation [41].

**Acknowledgments**

The work of A. Nedić and C.A. Uribe in Sect. 5 is supported by the National Science Foundation under grant no. CPS 15-44953. The research by P. Dvurechensky, D. Dvinskikh, and A. Gasnikov in Sect. 3 and Sect. 4 was funded by the Russian Science Foundation (project 18-71-10108).

## Footnotes

[1]The full version of this paper can be found in the supplementary material and is also available as [15].

[2]Formally, the $\rho$-Wasserstein distance for $\rho \geq 1$ is $(\mathcal{W}_0(\mu, \nu))^{\frac{1}{\rho}}$ if $\mathcal{Y} = \mathcal{Z}$ and $c_i(y) = d^\rho(z_i, y)$, $d$ being a distance on $\mathcal{Y}$. For simplicity, we refer to (1) as regularized Wasserstein distance in a general situation since our algorithm does not rely on any specific choice of cost $c_i(y)$.

[3]For simplicity, we assume equal weights for each $\mathcal{W}_{\gamma, \mu_i}(p)$ and do not normalize the sum dividing by $m$. Our results can be directly generalized to the case of non-negative weights summing up to 1.

[4] In the experiments, we use $\frac{1}{M_{k+1}}\sum_{r=1}^{M_{k+1}} p_i([\bar{\boldsymbol{\lambda}}_{k+1}]_i, Y_r^i)$ instead of $p_i([\bar{\boldsymbol{\lambda}}_{k+1}]_i, Y_1^i)$, which does not change the statement of Theorem 2, but reduces the variance of $\hat{\mathrm{p}}_N$ in practice. Moreover, under mild assumptions, we can obtain high-probability analogue to inequalities (9).

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
