[Supplementary Material]

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

- We provide iteration and arithmetic operations complexity for the proposed algorithms in terms of the problem parameters.

- We demonstrate the effectiveness of our algorithm on the distributed computation of the regularized Wasserstein barycenter of a set of Gaussian distributions and a set of von Mises distributions for various network topologies and network sizes. Moreover, we show some initial results on the problem of image aggregation for two datasets, namely, a subset of the MNIST digit dataset [31] and subset of the IXI Magnetic Resonance dataset [1].

## 1.3 Paper organization

This paper is organized as follows. In Section 2, we present the regularized Wasserstein barycenter problem for the semi-discrete case and its distributed computation over networks. In Section 3, we introduce a new algorithm for general stochastic optimization problems with linear constraints and analyze its convergence rate. Section 4 extends this algorithm and introduces our method for the distributed computation of regularized Wasserstein barycenter. Section 5 shows the experimental results for the proposed algorithm. The appendix contains the proofs of stated lemmas and theorems, as well as additional results of numerical experiments.

**Notation.** We define $\mathcal{M}_+^1(\mathcal{X})$ – the set of positive Radon probability measures on a metric space $\mathcal{X}$, and $S_1(n) = \{a \in \mathbb{R}_+^n \mid \sum_{l=1}^n a_l = 1\}$ the probability simplex. We use $\mathcal{C}(\mathcal{X})$ as the space of continuous functions on $\mathcal{X}$. We denote by $\delta(x)$ the Dirac measure at point $x$. We refer to $\lambda_{\max}(W)$ as the maximum eigenvalue of matrix W. We also use bold symbols for stacked vectors $\mathbf{p} = [p_1^T, \cdots, p_m^T]^T \in \mathbb{R}^{mn}$, where $p_1, ..., p_m \in \mathbb{R}^n$. In this case $[\mathbf{p}]_i = p_i$ – the $i$-th block of p. For a vector $\lambda \in \mathbb{R}^n$, we denote by $[\lambda]_l$ its $l$-th component. We refer to the Euclidean norm of a vector $\|p\|_2 := \sum_{l=1}^n ([p]_l)^2$ as 2-norm.

## 2 The Distributed Wasserstein Barycenter Problem

In this section, we present the problem of decentralized distributed computation of regularized Wasserstein barycenters for a family of possibly continuous probability measures distributed over a network. First, we provide the necessary background for entropic regularization of optimal transport and the Wasserstein barycenter problem. Then, we give the details of the distributed formulation of the optimization problem defining the Wasserstein barycenter, which is a minimization problem with linear equality constraint. To deal with this constraints, we make a transition to the dual problem, which, as we show, due to the presence of continuous distributions, is a smooth stochastic optimization problem.

## 2.1 Regularized semi-discrete formulation of the optimal transport problem

We consider entropic regularization for the optimal transport problem and the corresponding regularized Wasserstein distance and barycenter [12]. Let $\mu \in \mathcal{M}_+^1(\mathcal{Y})$ with density $q(y)$ on a metric space $\mathcal{Y}$, and a discrete probability measure $\nu = \sum_{i=1}^n [p]_i \delta(z_i)$ with weights given by vector $p \in S_1(n)$ and finite support given by points $z_1, \ldots, z_n \in \mathcal{Z}$ from a metric space $\mathcal{Z}$. Denote by $c_i(y) = c(z_i, y)$ a cost function for transportation of a unit of mass from point $z_i \in \mathcal{Z}$ to point $y \in \mathcal{Y}$. Then we define regularized Wasserstein distance in semi-discrete setting between continuous measure $\mu$ and discrete measure $\nu$ as follows[1]

$$\mathcal{W}_\gamma(\mu, \nu) = \min_{\pi \in \Pi(\mu,\nu)} \left\{ \sum_{i=1}^n \int_{\mathcal{Y}} c_i(y)\pi_i(y)dy + \gamma \sum_{i=1}^n \int_{\mathcal{Y}} \pi_i(y) \log\left( \frac{\pi_i(y)}{\xi} \right) dy \right\}, \qquad (1)$$

where $\xi$ is the uniform distribution on $\mathcal{Y} \times \mathcal{Z}$, and the set of admissible coupling measures $\pi$ is defined as follows

$$\Pi(\mu, \nu) = \left\{ \pi \in \mathcal{M}_+^1(\mathcal{Y}) \times S_1(n) : \sum_{i=1}^n \pi_i(y) = q(y), y \in \mathcal{Y}, \int_{\mathcal{Y}} \pi_i(y)dy = p_i, \forall\, i = 1, \ldots, n \right\}.$$

We emphasize that, unlike [22], we regularize the problem by the Kullback-Leibler divergence from the uniform distribution $\xi$, which allows us to find explicitly the Fenchel conjugate for $\mathcal{W}_\gamma(\mu, \nu)$, see Lemma 1 below.

For a set of measures $\mu_i \in \mathcal{M}_+^1(\mathcal{Z})$, $i = 1, \ldots, m$, we fix the support $z_1, \ldots, z_n \in \mathcal{Z}$ of their regularized Wasserstein barycenter $\nu$ and wish to find it in the form $\nu = \sum_{i=1}^n [p]_i \delta(z_i)$, where $p \in S_n(1)$. Then the regularized Wasserstein barycenter in the semi-discrete setting is defined as the solution to the following convex optimization problem[2]

$$\min_{p \in S_1(n)} \sum_{i=1}^m \mathcal{W}_{\gamma,\mu_i}(p), \qquad (2)$$

where we used notation $\mathcal{W}_{\gamma,\mu}(p) := \mathcal{W}_\gamma(\mu, \nu)$ for fixed probability measure $\mu$.

## 2.2 Network constraints in the distributed barycenter problem

We now describe the distributed optimization setting for solving problem (2). To do so, we rewrite the problem (2) in an equivalent form

$$\min_{\substack{p_1 = \cdots = p_m \\ p_1, \ldots, p_m \in S_1(n)}} \sum_{i=1}^m \mathcal{W}_{\gamma,\mu_i}(p_i). \qquad (3)$$

We assume that each measure $\mu_i$ is held by an agent $i$ on a network and this agent can sample from this measure. We model such a network as a fixed *connected undirected graph* $\mathcal{G} = (V, E)$, where $V$ is the set of $m$ nodes and $E$ is the set of edges. We assume that the graph $\mathcal{G}$ does not have self-loops. The network structure imposes information constraints, specifically, each node $i$ has access to $\mu_i$ only and can exchange information only with its immediate neighbors, i.e. nodes $j$ s.t. $(i, j) \in E$.

We represent the communication constraints imposed by the network by introducing a single equality constraint instead of the constraints $p_1 = \cdots = p_m$ in (3). To do so, we define the Laplacian matrix $\bar{W} \in \mathbb{R}^{m \times m}$ of the graph $\mathcal{G}$ as

$$[\bar{W}]_{ij} = \begin{cases} -1, & \text{if } (i,j) \in E, \\ \deg(i), & \text{if } i = j, \\ 0, & \text{otherwise}, \end{cases}$$

where $\deg(i)$ is the degree of the node $i$, i.e., the number of neighbors of the node. Finally, we define the communication matrix (also referred to as an interaction matrix) by $W := \bar{W} \otimes I_n$, where $\otimes$ denotes the Kronecker product of matrices.

Since the graph $\mathcal{G}$ is undirected and connected, the Laplacian matrix $\bar{W}$ is symmetric and positive semidefinite. Furthermore, the vector $\mathbf{1}$ of all ones is the unique (up to a scaling factor) eigenvector associated with the zero eigenvalue. In respect that the matrix $W$ inherits the properties of $\bar{W}$, i.e., it is symmetric and positive, we conclude that

$$W\mathbf{p} = 0 \text{ if and only if } p_1 = \cdots = p_m,$$

where $\mathbf{p} = [p_1^T, \cdots, p_m^T]^T \in \mathbb{R}^{mn}$. Moreover, this identity holds for $\sqrt{W} := \sqrt{\bar{W}} \otimes I_n$, i.e.

$$\sqrt{W}\mathbf{p} = 0 \text{ if and only if } p_1 = \cdots = p_m.$$

Using this fact, we equivalently rewrite problem (2) as the maximization problem with linear equality constraint

$$\max_{\substack{p_1,\ldots,p_m \in S_1(n) \\ \sqrt{W}\mathbf{p}=0}} -\sum_{i=1}^{m} \mathcal{W}_{\gamma,\mu_i}(p_i). \tag{4}$$

## 2.3 Dual formulation of the barycenter problem

Given that problem (4) is an optimization problem with linear constraints, we introduce a vector of dual variables $\boldsymbol{\lambda} = [\lambda_1^T, \cdots, \lambda_m^T]^T \in \mathbb{R}^{mn}$ for the constraints $\sqrt{W}\mathbf{p} = 0$ in (4). Then, the Lagrangian dual problem for (4) is

$$\min_{\boldsymbol{\lambda} \in \mathbb{R}^{mn}} \max_{p_1,\ldots,p_m \in S_1(n)} \left\{ \sum_{i=1}^{m} \langle \lambda_i, [\sqrt{W}\mathbf{p}]_i \rangle - \mathcal{W}_{\gamma,\mu_i}(p_i) \right\} = \min_{\boldsymbol{\lambda} \in \mathbb{R}^{mn}} \sum_{i=1}^{m} \mathcal{W}_{\gamma,\mu_i}^*([\sqrt{W}\boldsymbol{\lambda}]_i), \tag{5}$$

where $[\sqrt{W}\mathbf{p}]_i$ and $[\sqrt{W}\boldsymbol{\lambda}]_i$ denote the $i$-th $n$-dimensional block of vectors $\sqrt{W}\mathbf{p}$ and $\sqrt{W}\boldsymbol{\lambda}$ respectively, the equality $\sum_{i=1}^{m} \langle \lambda_i, [\sqrt{W}\mathbf{p}]_i \rangle = \sum_{i=1}^{m} \langle [\sqrt{W}\boldsymbol{\lambda}]_i, p_i \rangle$ was used, and $\mathcal{W}_{\gamma,\mu_i}^*(\cdot)$ is the Fenchel-Legendre transform of $\mathcal{W}_{\gamma,\mu_i}(p_i)$. The following Lemma states that each $\mathcal{W}_{\gamma,\mu_i}^*(\cdot)$ is a smooth function with Lipschitz-continuous gradient and can be expressed as an expectation of a function of additional random argument.

**Lemma 1.** *Given a positive Radon probability measure $\mu \in \mathcal{M}_+^1(\mathcal{Y})$ with density $q(y)$ on a metric space $\mathcal{Y}$, the Fenchel-Legendre dual function for $\mathcal{W}_{\gamma,\mu}(p)$ has the following explicit form*

$$\mathcal{W}_{\gamma,\mu}^*(\bar{\lambda}) = \mathbb{E}_{Y \sim \mu} \gamma \log \left( \frac{1}{q(Y)} \sum_{\ell=1}^{n} \exp \left( \frac{[\bar{\lambda}]_\ell - c_\ell(Y)}{\gamma} \right) \right),$$

*and its gradient is $1/\gamma$-Lipschitz continuous w.r.t. 2-norm with following l-th component*

$$[\nabla \mathcal{W}_{\gamma,\mu}^*(\bar{\lambda})]_l = \mathbb{E}_{Y \sim \mu} \frac{\exp(([\bar{\lambda}]_l - c_l(Y))/\gamma)}{\sum_{\ell=1}^{n} \exp(([\bar{\lambda}]_\ell - c_\ell(Y))/\gamma)}, \ l = 1,\ldots,n,$$

*where $Y \sim \mu$ means that random variable $Y$ is distributed according to measure $\mu$.*

Denote $\bar{\boldsymbol{\lambda}} = \sqrt{W}\boldsymbol{\lambda} = [[\sqrt{W}\boldsymbol{\lambda}]_1^T, \ldots, [\sqrt{W}\boldsymbol{\lambda}]_m^T]^T = [\bar{\lambda}_1^T, \ldots, \bar{\lambda}_m^T]^T$ and $\mathcal{W}_\gamma^*(\boldsymbol{\lambda})$ – the dual objective in the r.h.s. of (5). Then, by the chain rule, the $l$-th $n$-dimensional block of $\nabla \mathcal{W}_\gamma^*(\boldsymbol{\lambda})$ is

$$\left[ \nabla \mathcal{W}_\gamma^*(\boldsymbol{\lambda}) \right]_l = \left[ \nabla \sum_{i=1}^{m} \mathcal{W}_{\gamma,\mu_i}^*([\sqrt{W}\boldsymbol{\lambda}]_i) \right]_l = \sum_{j=1}^{m} \sqrt{W}_{lj} \nabla \mathcal{W}_{\gamma,\mu_j}^*(\bar{\lambda}_j), \ l = 1,...,m. \tag{6}$$

From Lemma 1 and the expression (6) for the gradient of the dual objective, we can see that the *dual* problem (5) is a smooth stochastic convex optimization problem. This is in contrast to [30], where the primal problem is a stochastic optimization problem. Moreover, as opposed to the existing literature on stochastic convex optimization, we not only need to solve the dual problem but also need to reconstruct an approximate solution for the primal problem (4), which is the barycenter. In order to do this, in the next section, we develop a novel accelerated primal-dual stochastic gradient method for a general smooth stochastic optimization problem, which is dual to some optimization problem with linear equality constraints. Further, in Section 4, we apply our general algorithm to the particular case of primal-dual pair of problems (4) and (5).

# 3 General Primal-Dual Framework for Stochastic Optimization

In this section, we consider a general smooth stochastic convex optimization problem which is dual to some optimization problem with linear equality constraints. Extending our works [16, 21, 10, 17, 19, 3, 18], we develop a novel algorithm for its solution and reconstruction of the primal variable together with convergence rate analysis. We underline that the material of this section is not standard. Unlike prior works, we consider the stochastic primal-dual pair of problems and one of our contributions consists in providing a primal-dual extension of the accelerated stochastic gradient method. We believe that our algorithm can be used for problems other than regularized Wasserstein barycenter problem and, thus, we, first, provide a general algorithm and, then, apply it to the barycenter problem. We introduce new notation since this section is independent of the others and is focused on a general optimization problem.

## 3.1 General setup and assumptions

For any finite-dimensional real vector space $E$, we denote by $E^*$ its dual, by $\langle \lambda, x \rangle$ the value of a linear function $\lambda \in E^*$ at $x \in E$. Let $\|\cdot\|$ denote some norm on $E$ and $\|\cdot\|_*$ denote the norm on $E^*$ which is dual to $\|\cdot\|$, i.e. $\|\lambda\|_* = \max\{\langle \lambda, x \rangle : \|x\| \leq 1\}$. For a linear operator $A : E_1 \to E_2$, we define the adjoint operator $A^T : E_2^* \to E_1^*$ in the following way $\langle u, Ax \rangle = \langle A^T u, x \rangle$, $\forall u \in E_2^*, x \in E_1$. We say that a function $f : E \to \mathbb{R}$ has a $L$-Lipschitz continuous gradient w.r.t. norm $\|\cdot\|_*$ if it is continuously differentiable and its gradient satisfies Lipschitz condition $\|\nabla f(x) - \nabla f(y)\|_* \leq L\|x - y\|$, $\forall x, y \in E$. Note that, from this inequality, it follows that

$$f(y) \leq f(x) + \langle \nabla f(x), y - x \rangle + \frac{L}{2}\|x - y\|^2, \quad \forall x, y \in E. \tag{7}$$

The main problem, we consider in this section, is a where $Q$ is a simple closed convex set, $A : E \to H$ is given linear operator, $b \in H$ is given, $\Lambda = H^*$. We define

$$\varphi(\lambda) := \langle \lambda, b \rangle + \max_{x \in Q} \left( -f(x) - \langle A^T \lambda, x \rangle \right) = \langle \lambda, b \rangle + f^*(-A^T \lambda) \tag{8}$$

and assume it to be smooth with $L$-Lipschitz continuous gradient. Here $f^*$ is the Fenchel-Legendre conjugate function for $f$. We also assume that $f^*(-A^T\lambda) = \mathbb{E}_\xi F^*(-A^T\lambda, \xi)$, where $\xi$ is random vector and $F^*$ is the Fenchel-Legendre conjugate function to some function $F(x, \xi)$, i.e. it satisfies $F^*(-A^T\lambda, \xi) = \max_{x \in Q}\{\langle -A^T\lambda, x \rangle - F(x, \xi)\}$. $F^*(\bar{\lambda}, \xi)$ is assumed to be smooth and, hence $\nabla_{\bar{\lambda}} F^*(\bar{\lambda}, \xi) = x(\bar{\lambda}, \xi)$, where $x(\bar{\lambda}, \xi)$ is the solution of the maximization problem

$$x(\bar{\lambda}, \xi) = \arg \max_{x \in Q}\{\langle \bar{\lambda}, x \rangle - F(x, \xi)\}.$$

Further, we assume that the dual problem $(D)$ can be accessed by a stochastic oracle $(\Phi(\lambda, \xi), \nabla\Phi(\lambda, \xi))$ with $\Phi(\lambda, \xi) = \langle \lambda, b \rangle + F^*(-A^T\lambda, \xi)$ and $\nabla\Phi(\lambda, \xi) = b - A\nabla F^*(-A^T\lambda, \xi)$ satisfying

$$\mathbb{E}_\xi \Phi(\lambda, \xi) = \varphi(\lambda), \quad \mathbb{E}_\xi \nabla\Phi(\lambda, \xi) = \nabla\varphi(\lambda), \quad \mathbb{E}_\xi \|\nabla\Phi(\lambda, \xi) - \nabla\varphi(\lambda)\|_2^2 \leq \sigma^2, \lambda \in H^*. \tag{9}$$

Finally, we assume that dual problem $(D)$ has a solution $\lambda^*$ and there exists some $R > 0$ such that $\|\lambda^*\|_2 \leq R < +\infty$.

## 3.2 An accelerated stochastic gradient method

To solve the primal-dual pair of problems $(P) - (D)$, our first step, which we do in this subsection, is to introduce and analyse an accelerated stochastic gradient method (see Algorithm 1) for a general stochastic optimization problem and obtain some basic properties of the generated sequences, see Theorem 1. In the next subsection, we apply it to the dual problem $(D)$. Algorithm 1 is close in its form to the one in [29], but we use a different analysis extending [19] for the stochastic case.

To describe our algorithm, we introduce *proximal setup*, which is usually used in proximal gradient methods, see e.g. [6]. We choose some norm $\|\cdot\|$ on the space of vectors $\lambda$ and a *prox-function* $d(\lambda) : \Lambda \to \mathbb{R}$ which is convex, continuous on $\Lambda$, continuously differentiable and 1-strongly convex on $\Lambda_0 = \{\lambda \in \Lambda : \partial d(\lambda) \neq \emptyset\}$ with respect to $\|\cdot\|$, i.e., $\forall \lambda \in \Lambda, \zeta \in \Lambda^0$ $d(\lambda) - d(\zeta) - \langle \nabla d(\zeta), \lambda -$

$\zeta\rangle \geq \frac{1}{2}\|\lambda - \zeta\|^2$. Here $\partial d(\lambda)$ is the subdifferential of $d$ and $\nabla d(x)$ is its subgradient. We define also the corresponding *Bregman divergence* $V[\zeta](\lambda) := d(\lambda) - d(\zeta) - \langle \nabla d(\zeta), \lambda - \zeta \rangle, \lambda \in \Lambda, \zeta \in \Lambda^0$. It is easy to see that

$$V[\zeta](\lambda) \geq \frac{1}{2}\|\lambda - \zeta\|^2, \quad \forall \, \lambda \in \Lambda, \zeta \in \Lambda^0. \tag{10}$$

---

**Algorithm 1** Accelerated Stochastic Gradient Method (ASGD)

---

**Input:** Starting point $\lambda_0 \in \Lambda$, prox-setup: $d(\lambda) - 1$-strongly convex w.r.t. $\|\cdot\|$, the number of iterations $N$, Bregman divergence $V[\zeta](\lambda) := d(\lambda) - d(\zeta) - \langle \nabla d(\zeta), \lambda - \zeta \rangle, \lambda \in \Lambda, \zeta \in \Lambda^0$.

1: $C_0 = \alpha_0 = 0, \eta_0 = \zeta_0 = \lambda_0$.
2: **for** $k = 0, \ldots, N-1$ **do**
3:     Find $\alpha_{k+1}$ as the largest root of the equation

$$C_{k+1} := C_k + \alpha_{k+1} = 2L\alpha_{k+1}^2. \tag{11}$$

4:

$$\lambda_{k+1} = \frac{\alpha_{k+1}\zeta_k + C_k\eta_k}{C_{k+1}}. \tag{12}$$

5:

$$\zeta_{k+1} = \arg\min_{\lambda \in \Lambda}\{V[\zeta_k](\lambda) + \alpha_{k+1}(\Phi(\lambda_{k+1}, \xi_{k+1}) + \langle \nabla\Phi(\lambda_{k+1}, \xi_{k+1}), \lambda - \lambda_{k+1} \rangle)\}. \tag{13}$$

6:

$$\eta_{k+1} = \frac{\alpha_{k+1}\zeta_{k+1} + C_k\eta_k}{C_{k+1}}. \tag{14}$$

7: **end for**
**Output:** The point $\eta_N$.

---

**Theorem 1.** *Let the sequences* $\{\lambda_N, \eta_N, \zeta_N, \alpha_N, C_N\}$, $N > 0$ *be generated by Algorithm 1. Then, for all $N > 0$, it holds that*

$$C_N\varphi(\eta_N) \leq \min_{\lambda \in \Lambda}\left\{\sum_{k=0}^{N}\alpha_k\left(\varphi(\lambda_k) + \langle \nabla\Phi(\lambda_k, \xi_k), \lambda - \lambda_k \rangle\right) + V[\zeta_0](\lambda)\right\}$$
$$+ \sum_{k=0}^{N-1}C_{k+1}\langle \nabla\Phi(\lambda_{k+1}, \xi_{k+1}) - \nabla\varphi(\lambda_{k+1}), \eta_k - \lambda_{k+1} \rangle + \sum_{k=0}^{N}\frac{C_k}{2L}\|\nabla\Phi(\lambda_k, \xi_k) - \nabla\varphi(\lambda_k)\|_*^2. \tag{15}$$

## 3.3 Accelerated primal-dual stochastic gradient method

In this subsection, we develop an accelerated algorithm for the primal-dual pair of problems $(P)-(D)$. The idea is to apply the algorithm of the previous subsection to the dual problem $(D)$, endow it with a step in the primal space and, using the result of Theorem 1, show that the new algorithm allows to approximate also the solution to the primal problem. Since the feasible set in the problem $(D)$ is unbounded, we choose the Euclidean proximal setup in $H^*$ and denote the standard Euclidean norm by $\|\cdot\|_2$. We use Euclidean proximal setup with the prox-function $d(\lambda) = \frac{1}{2}\|\lambda\|_2^2$ and the Bregman divergence $V[\zeta](\lambda) = \frac{1}{2}\|\lambda - \zeta\|_2^2$.

Note that, in this case, the dual norm is also Euclidean and the step 5 of the algorithm simplifies. We additionally assume that the variance of the stochastic approximation $\nabla\Phi(\lambda, \xi)$ for the gradient of $\varphi$ can be controlled and made as small as we desire. This can be done, for example by mini-batching the stochastic approximation. Finally, since $\nabla\Phi(\lambda, \xi) = b - A\nabla F^*(-A^T\lambda, \xi) = b - Ax(-A^T\lambda, \xi)$, on each iteration, to find $\nabla\Phi(\lambda, \xi)$ we find the vector $x(-A^T\lambda, \xi)$ and use it for the primal iterates.

**Theorem 2.** *Let $\varphi$ have $L$-Lipschitz continuous gradient w.r.t. 2-norm and $\|\lambda^*\|_2 \leq R$, where $\lambda^*$ is a solution of dual problem $(D)$. Assume that at each iteration of Algorithm 2, the stochastic approximation $\nabla\Phi(\lambda_k, \xi_k)$ of the gradient is chosen in such a way that $\mathbb{E}_\xi\|\nabla\Phi(\lambda_k, \xi_k) - \nabla\varphi(\lambda_k)\|_2^2 \leq \frac{\varepsilon L\alpha_k}{C_k}$. Then, for any $\varepsilon > 0$ and $N \geq 0$, the output $\hat{x}_N$ generated by the Algorithm 2 satisfies*

$$f(\mathbb{E}\hat{x}_N) - f^* \leq \frac{16LR^2}{N^2} + \frac{\varepsilon}{2} \quad \text{and} \quad \|A\mathbb{E}\hat{x}_N - b\|_2 \leq \frac{16LR}{N^2} + \frac{\varepsilon}{2R}, \tag{20}$$

*where the expectation is taken w.r.t. all the randomness $\xi_1, \ldots, \xi_N$.*

**Algorithm 2** Accelerated Primal-Dual Stochastic Gradient Method (APDSGD)

---

**Input:** starting point $\lambda_0 = 0$, the number of iterations $N$.

1: $C_0 = \alpha_0 = 0$, $\eta_0 = \zeta_0 = \lambda_0 = \hat{x}_0 = 0$.
2: **for** $k = 0, \dots, N-1$ **do**
3:    Find $\alpha_{k+1}$ as the largest root of the equation

$$C_{k+1} := C_k + \alpha_{k+1} = 2L\alpha_{k+1}^2. \tag{16}$$

4:

$$\lambda_{k+1} = \frac{\alpha_{k+1}\zeta_k + C_k\eta_k}{C_{k+1}}. \tag{17}$$

5:

$$\zeta_{k+1} = \zeta_k - \alpha_{k+1}\nabla\Phi(\lambda_{k+1}, \xi_{k+1}). \tag{18}$$

6:

$$\eta_{k+1} = \frac{\alpha_{k+1}\zeta_{k+1} + C_k\eta_k}{C_{k+1}}. \tag{19}$$

7:    Set

$$\hat{x}_{k+1} = \frac{1}{C_{k+1}}\sum_{i=0}^{k+1} \alpha_i x(-A^T\lambda_i, \xi_i) = \frac{\alpha_{k+1}x(-A^T\lambda_{k+1}, \xi_{k+1}) + C_k\hat{x}_k}{C_{k+1}}.$$

8: **end for**
**Output:** The points $\hat{x}_N, \eta_N$.

---

In step 7 of Algorithm 2 we can use a batch of size $M$ and $\frac{1}{M}\sum_{r=1}^M x(\lambda_{k+1}, \xi_{k+1}^r)$ to update $\hat{x}_{k+1}$. Then, under reasonable assumptions, $\hat{x}_N$ concentrates around $\mathbb{E}\hat{x}_N$ [23] and, if $f$ is Lipschitz-continuous, we obtain that (20) holds with large probability with $\hat{x}_N$ instead of $\mathbb{E}\hat{x}_N$.

## 4  Solving the Barycenter Problem

In this section, we apply the general algorithm APDSGD from the previous section to solve the primal-dual pair of problems (4)-(5) and approximate the regularized Wasserstein barycenter which is a solution to (4). First, in Lemma 2, we make a number of technical steps to take care of the assumptions of Theorem 2. We estimate the Lipschitz constant of the dual objective's gradient in (5), introduce mini-batch stochastic approximation for the gradient of the dual objective and estimate its variance. Then, we introduce a change of dual variable so that a gradient-type step for the dual objective, e.g., the step 5 of Algorithm 2, becomes feasible for the decentralized distributed setting. Then, for simplicity, we consider a non-accelerated algorithm for regularized Wasserstein barycenter problem to illustrate the combination of gradient methods, a stochastic approximation of the gradient and decentralized distributed computations. Finally, we present our accelerated algorithm for regularized Wasserstein barycenter problem with its complexity analysis.

**Lemma 2.** *The gradient of the dual objective function $\mathcal{W}_\gamma^*(\boldsymbol{\lambda})$ in the dual problem (5) is $\lambda_{\max}(W)/\gamma$-Lipschitz continuous w.r.t. 2-norm. If its stochastic approximation is defined as*

$$[\widetilde{\nabla}\mathcal{W}_\gamma^*(\boldsymbol{\lambda})]_i = \sum_{j=1}^m \sqrt{W}_{ij}\widetilde{\nabla}\mathcal{W}_{\gamma,\mu_j}^*(\bar{\lambda}_j), \ i = 1, \dots, m, \quad with \tag{21}$$

$$\widetilde{\nabla}\mathcal{W}_{\gamma,\mu_j}^*(\bar{\lambda}_j) = \frac{1}{M}\sum_{r=1}^M p_j(\bar{\lambda}_j, Y_r^j), \ j = 1, \dots, m, \quad and \tag{22}$$

$$[p_j(\bar{\lambda}_j, Y_r^j)]_l = \frac{\exp(([\bar{\lambda}_j]_l - c_l(Y_r^j))/\gamma)}{\sum_{\ell=1}^n \exp(([\bar{\lambda}_j]_\ell - c_\ell(Y_r^j))/\gamma)}, \ j = 1, \dots, m, \ l = 1, \dots, n, \ r = 1, \dots, M \tag{23}$$

*where $M$ is the batch size, $Y_1^j, \dots, Y_r^j$ is a sample from the measures $\mu_j$, $j = 1, \dots, m$. Then*

$$\mathbb{E}\widetilde{\nabla}\mathcal{W}_\gamma^*(\boldsymbol{\lambda}) = \nabla\mathcal{W}_\gamma^*(\boldsymbol{\lambda}) \quad and$$

$$\mathbb{E}\|\widetilde{\nabla}\mathcal{W}_\gamma^*(\boldsymbol{\lambda}) - \nabla\mathcal{W}_\gamma^*(\boldsymbol{\lambda})\|_2^2 \le \frac{\lambda_{\max}(W)m}{M}, \ \boldsymbol{\lambda} \in \mathbb{R}^{mn},$$

*where the expectation is taken w.r.t. all samples $(Y_1^j, \ldots Y_M^j)$ from measure $\mu_j$, $j = 1, \ldots, m$.*

Let us consider a simple stochastic gradient step for the particular dual problem (5). Note that the step 5 of Algorithm 2 has the same form. Using (6), the stochastic gradient step $\boldsymbol{\lambda}_{k+1} = \boldsymbol{\lambda}_k - \frac{1}{L}\widetilde{\nabla}\mathcal{W}_\gamma^*(\boldsymbol{\lambda}_k)$ can be written block-wise as

$$[\boldsymbol{\lambda}_{k+1}]_i = [\boldsymbol{\lambda}_k]_i - \frac{1}{L}\sum_{j=1}^m \sqrt{W}_{ij}\widetilde{\nabla}\mathcal{W}_{\gamma,\mu_j}^*([\sqrt{W}\boldsymbol{\lambda}_k]_j), \quad \text{for each agent } i = 1, ..., m,$$

where $\widetilde{\nabla}\mathcal{W}_{\gamma,\mu_j}^*(\cdot)$ is defined in (22) with the batch size $M = 1$, and $L = \lambda_{\max}(W)/\gamma$. Unfortunately, this update can not be made in the decentralized setting since the sparsity pattern of $\sqrt{W}_{ij}$ can be different from $W_{ij}$ and this will require some agents to get information not only from their neighbors. To overcome this obstacle, we change the variable and denote $\bar{\boldsymbol{\lambda}} = \sqrt{W}\boldsymbol{\lambda}$. Then the gradient step becomes

$$[\bar{\boldsymbol{\lambda}}_{k+1}]_i = [\bar{\boldsymbol{\lambda}}_k]_i - \frac{1}{L}\sum_{j=1}^m W_{ij}\widetilde{\nabla}\mathcal{W}_{\gamma,\mu_j}^*([\bar{\boldsymbol{\lambda}}_k]_j), \quad \text{for each agent } i = 1, ..., m.$$

Algorithm 3 presents a non-accelerated primal-dual stochastic gradient method, combining distributed updates and stochastic gradient step described above. This algorithm solves the primal-dual pair of problems (4)-(5) and approximates the regularized Wasserstein barycenter which is a solution to (4). The algorithm has a loop, indexed by iteration number $k$ and the index $i$ corresponds to the agent's number. At each iteration $k$ of the algorithm, each agent $i$ samples from the measure $\mu_i$ and forms a stochastic approximation of the gradient of $\mathcal{W}_{\gamma,\mu_i}(\cdot)$. Then each agent shares this vector with its neighbors. After that, each agent calculates a step direction based on its information and information gathered from the neighbors. Note that the matrix $W$ provides communications only between neighboring nodes and step 6 requires only local information.

---

**Algorithm 3** Non-accelerated Distributed Computation of Wasserstein barycenter

---

**Input:** Each agent $i \in V$ is assigned its measure $\mu_i$.
1: All agents set $[\bar{\boldsymbol{\lambda}}_0]_i = \mathbf{0} \in \mathbb{R}^n$, $[\hat{\mathrm{p}}_0]_i = \mathbf{0} \in \mathbb{R}^n$, and $N$.
2: For each agent $i \in V$:
3: **for** $k = 0, \ldots, N-1$ **do**
4:     Sample $Y^i$ from the measure $\mu_i$ and set $\widetilde{\nabla}\mathcal{W}_{\gamma,\mu_i}^*([\bar{\boldsymbol{\lambda}}_k]_i)$ as defined in (22) with $M = 1$.
5:     Share $\widetilde{\nabla}\mathcal{W}_{\gamma,\mu_i}^*([\bar{\boldsymbol{\lambda}}_k]_i)$ with $\{j \mid (i,j) \in E\}$
6:

$$[\bar{\boldsymbol{\lambda}}_{k+1}]_i = [\bar{\boldsymbol{\lambda}}_k]_i - \frac{1}{L}\sum_{j=1}^m W_{ij}\widetilde{\nabla}\mathcal{W}_{\gamma,\mu_j}^*([\bar{\boldsymbol{\lambda}}_k]_j).$$

7:     $[\hat{\mathrm{p}}_{k+1}]_i = [\hat{\mathrm{p}}_k]_i + \frac{1}{N}p_i([\bar{\boldsymbol{\lambda}}_{k+1}]_i, Y^i)$, where $p_i(\cdot, \cdot)$ is defined in (23).
8: **end for**
**Output:** $\hat{\mathrm{p}}_N$.

---

Finally, we apply accelerated primal-dual stochastic gradient method (APDSGD) from the previous section to solve the primal-dual pair of problems (4)-(5) and calculate the regularized Wasserstein barycenter. As above, we introduce the change of dual variables $\bar{\boldsymbol{\lambda}} = \sqrt{W}\boldsymbol{\lambda}$, $\bar{\boldsymbol{\eta}} = \sqrt{W}\boldsymbol{\eta}$, $\bar{\boldsymbol{\zeta}} = \sqrt{W}\boldsymbol{\zeta}$, which makes the step 5 of Algorithm 2 feasible for the decentralized distributed setting. The result is Algorithm 4. At each iteration $k$ each agent $i$ generates a sample of size $M_k$ from measure $\mu_i$, forms a stochastic approximation of the gradient of $\mathcal{W}_{\gamma,\mu_i}(\cdot)$ according to (22) and shares it with the neighbors. The mini-batch size $M_k$ is chosen such that $M_k \geq \frac{m\gamma C_k}{\alpha_k \varepsilon}$, which, by Lemma 2, means that $\mathbb{E}\|\widetilde{\nabla}\mathcal{W}_\gamma^*(\boldsymbol{\lambda}) - \nabla\mathcal{W}_\gamma^*(\boldsymbol{\lambda})\|_2^2 \leq \frac{\varepsilon L \alpha_k}{C_k}$ and the assumptions of Theorem 2 hold.

**Algorithm 4** Accelerated Distributed Computation of Wasserstein barycenter

**Input:** Each agent $i \in V$ is assigned its measure $\mu_i$.
1: All agents $i \in V$ set $[\bar{\boldsymbol{\eta}}_0]_i = [\bar{\boldsymbol{\zeta}}_0]_i = [\bar{\boldsymbol{\lambda}}_0]_i = [\hat{\mathrm{p}}_0]_i = \mathbf{0} \in \mathbb{R}^n$, $C_0 = \alpha_0 = 0$ and $N$.
2: For each agent $i \in V$:
3: **for** $k = 0, \dots, N-1$ **do**
4:     Find $\alpha_{k+1}$ as the largest root of the equation $C_{k+1} := C_k + \alpha_{k+1} = \frac{2\lambda_{\max}(W)\alpha_{k+1}^2}{\gamma}$.
5:
$$M_{k+1} = \max\left\{1, \left\lceil \frac{m\gamma C_{k+1}}{\alpha_{k+1}\varepsilon} \right\rceil\right\}.$$

6:
$$[\bar{\boldsymbol{\lambda}}_{k+1}]_i = \frac{\alpha_{k+1}[\bar{\boldsymbol{\zeta}}_k]_i + C_k[\bar{\boldsymbol{\eta}}_k]_i}{C_{k+1}}.$$

7:     Generate $M_{k+1}$ samples $\{Y_r^i\}_{r=1}^{M_{k+1}}$ from the measure $\mu_i$ and set $\widetilde{\nabla}\mathcal{W}_{\gamma,\mu_i}^*([\bar{\boldsymbol{\lambda}}_k]_i)$ as in (22) with $M = M_k$.
8:     Share $\widetilde{\nabla}\mathcal{W}_{\gamma,\mu_i}^*([\bar{\boldsymbol{\lambda}}_{k+1}]_i)$ with $\{j \mid (i,j) \in E\}$.
9:
$$[\bar{\boldsymbol{\zeta}}_{k+1}]_i = [\bar{\boldsymbol{\zeta}}_k]_i - \alpha_{k+1}\sum_{j=1}^{m} W_{ij}\widetilde{\nabla}\mathcal{W}_{\gamma,\mu_j}^*([\bar{\boldsymbol{\lambda}}_{k+1}]_j).$$

10:
$$[\bar{\boldsymbol{\eta}}_{k+1}]_i = \frac{\alpha_{k+1}[\bar{\boldsymbol{\zeta}}_{k+1}]_i + C_k[\bar{\boldsymbol{\eta}}_{k+1}]_i}{C_{k+1}}.$$

11:
$$[\hat{\mathrm{p}}_{k+1}]_i = \frac{1}{C_{k+1}}\sum_{i=0}^{k+1}\alpha_i p_i([\bar{\boldsymbol{\lambda}}_{k+1}]_i, Y_1^i) = \frac{\alpha_{k+1}p_i([\bar{\boldsymbol{\lambda}}_{k+1}]_i, Y_1^i) + C_k[\hat{\mathrm{p}}_k]_i}{C_{k+1}},$$

    where $p_i(\cdot, \cdot)$ is defined in (23). [3]
12: **end for**
**Output:** $\hat{\mathrm{p}}_N$.

**Theorem 3.** *Let the assumptions of Section 2 hold and $R$ be such that $\|\boldsymbol{\lambda}^*\|_2 \leq R$. Then Algorithm 4 after $N = \sqrt{32\lambda_{\max}(W)R^2/(\varepsilon\gamma)}$ iterations returns an approximation $\hat{\mathrm{p}}_N$ for the barycenter, which satisfies*

$$\sum_{i=1}^{m}\mathcal{W}_{\gamma,\mu_i}(\mathbb{E}[\hat{\mathrm{p}}_N]_i) - \sum_{i=1}^{m}\mathcal{W}_{\gamma,\mu_i}([\mathrm{p}^*]_i) \leq \varepsilon, \quad \|\sqrt{W}\mathbb{E}\hat{\mathrm{p}}_N\|_2 \leq \varepsilon/R. \qquad (24)$$

*Moreover, the total complexity is $O\left(mn\max\left\{\sqrt{\frac{\lambda_{\max}(W)R^2}{\varepsilon\gamma}}, \frac{\lambda_{\max}(W)mR^2}{\varepsilon^2}\right\}\right)$ arithmetic operations.*

We underline that even if the measures $\mu_i$, $i = 1, \dots, m$ are discrete with large support size, it can be more efficient to apply our stochastic algorithm than to apply a deterministic algorithm. We now explain it in more details. If a measure $\mu$ is discrete, then $\mathcal{W}_{\gamma,\mu}^*(\bar{\lambda})$ in Lemma 1 is represented as a finite expectation, i.e., is a sum of functions instead of an integral, and can be found explicitly. In the same way, its gradient and, hence, the gradient of the dual objective $\mathcal{W}_\gamma^*(\boldsymbol{\lambda})$ in (6) can be found explicitly in a deterministic way. Then a deterministic accelerated primal-dual decentralized algorithm can be applied to approximate the regularized barycenter. Let us assume for simplicity that the support of measure $\mu$ is of the size $n$. Then the calculation of the exact gradient of $\mathcal{W}_{\gamma,\mu}^*(\bar{\lambda})$ requires $O(n^2)$ arithmetic operations and the overall complexity of the deterministic algorithm is $O\left(mn^2\sqrt{\lambda_{\max}(W)R^2/\gamma\varepsilon}\right)$. For comparison, the complexity of our randomized approach in Theorem 3 is proportional to $n$, but not to $n^2$. So, our randomized approach is superior in the regime of large $n$.

Figure 1: Dual function value and distance to consensus for $200, 100, 10, 500$ agents, $M_k = 100$ and $\gamma = 0.1$.

It is also interesting to compare the complexity of the accelerated method in Theorem 3 with the complexity of non-accelerated Algorithm 3. Similarly to the proof of Theorem 2, extending the convergence rate proof of stochastic Mirror Descent [25] for the primal-dual pair of problems $(P) - (D)$, we obtain the complexity of the non-accelerated method to be $O\left(mn \max\left\{\lambda_{\max}(W)R^2/(\varepsilon\gamma),\ \lambda_{\max}(W)mR^2/\varepsilon^2\right\}\right)$. As we see, acceleration improves the dependence on the $\lambda_{\max}(W)R^2/(\varepsilon\gamma)$, which is important, for example, for the limiting case $\gamma \to 0$, corresponding to approximation of the non-regularized barycenter.

## 5   Experimental Results

In this section, we present experimental results for Algorithm 4. Initially, we consider a set of agents over a network, where each agent $i$ can query realizations (i.e., samples) from a privately held random variable $Y_i \sim \mathcal{N}(\theta_i, v_i^2)$, where $\mathcal{N}(\theta, v^2)$ is a univariate Gaussian distribution with mean $\theta$ and variance $v^2$. Moreover, we set $\theta_i \in [-4, 4]$ and $v_i \in [0.1, 0.6]$. The objective is to compute a discrete distribution $p \in S_1(n)$ that solves (2). We assume $n = 100$ and the support of $p$ is a set of 100 equally spaced points on the segment $[-5, 5]$. Figure 1 shows the performance of Algorithm 4 for four classes of networks: complete, cycle, star, and Erdős-Rényi. Moreover, we show the behavior for different network sizes, namely: $m = 10, 100, 200, 500$. Particularly we use two metrics: the function value of the dual problem and the distance to consensus, i.e., $\mathcal{W}_\gamma^*(\boldsymbol{\lambda}) = \sum_{i=1}^m \mathcal{W}_{\gamma,\mu_i}^*([\bar{\boldsymbol{\lambda}}]_i)$ and $C(\hat{p}) := \|\sqrt{W}\hat{p}\|_2$. As expected, when the network is a complete graph, the convergence to the final value and the distance to consensus decreases rapidly. Nevertheless, the performance in graphs with degree regularity, such as the cycle graph and the Erdős-Rényi random graph, is similar to a complete graph with much less communication overhead. For the star graph, which has the worst case between the maximum and minimum number of neighbors among all nodes, the algorithms performs poorly. The reason is that despite the diameter of the graph is 2, $\lambda_{\max}(W)$, which appears in the complexity bounds, is of the order of number of vertices $m$.

Figure 2(a) shows a sample of the local barycenters of 10 agents on an Erdős-Rényi random graph, with local Gaussian distributions, at different times of the Algorithm 4, $N = 1, 100, 200, 500$. The local barycenters of all the agents in the network converge to a common distribution. Similarly, Figure 2(b) shows the convergence of the local barycenters of the agents on the same Erdős-Rényi random graph when the local distributions are von Mises distributions. Particularly, for the cases of von Mises distributions, we have used the angle between to points distance function.

Figure 3 shows the computed local barycenter of 9 agents in a network of 500 nodes at different iteration numbers. Each agent holds a local copy of a sample of the digit 2 ($56 \times 56$ image) from the MNIST dataset [31]. All agents converge to the same image that structurally represents the aggregation of the original 500 images held over the network. Finally, Figure 4 shows a simple example of an application of Wasserstein barycenter on medical image aggregation where we have 4

(a) Local Gaussian distributions

(b) Local von Mises distributions

Figure 2: Local barycenter of a set of Gaussian distribution and von Mises distributions. Barycenter is generated by the Algorithm 4 for a set of 10 agents over an Erdős-Rényi random graph at different iteration numbers. Each agent can access private realizations from a von Mises random variable.

| $N = 1$ | $N = 1000$ | $N = 2000$ | $N = 3000$ | $N = 4000$ |

Figure 3: Wasserstein barycenter of a subset of images of the digit 2 from the MNIST dataset [31]. Each block shows a subset of 9 randomly selected local barycenters, generated by Algorithm 4 at different time instances. The 9 agents are selected from a network of 500 agents on an Erdős-Rényi random graph.

| $N = 1$ | $N = 100$ | $N = 1000$ | $N = 6000$ | $N = 10000$ |

Figure 4: Wasserstein barycenter for a subset of images from the IXI dataset [1]. Each block shows the local barycenters, of 4 agents, generated by Algorithm 4 at different time instances. The 4 agents are connected on a cycle graph.

agents connected over a cycle graph and each agent holds a magnetic resonance image ($256 \times 256$) from the IXI dataset [1].

## 6 Conclusions and Future Directions

We propose a novel distributed algorithm for the computation of the regularized Wasserstein barycenter of a set of continuous measures stored distributedly over a network of agents. Moreover, we provide explicit and non-asymptotic iteration and sample complexity analysis in terms of the problem parameters and the network topology. Our algorithm is based on a new general algorithm for the solution of stochastic convex optimization problems with linear constraints. In contrast to the recent literature, our algorithm can be executed over arbitrary connected and static networks where nodes are oblivious to the network topology, which makes it suitable for large-scale network optimization set-

ting. Additionally, our analysis indicates that the randomization strategy provides faster convergence rates than the deterministic procedure when the support size of the barycenter is large.

The presented experiments were carried out in a single machine and implementation of our algorithm on real networks is a major research thrust for future projects. Extending fast distributed algorithms for the case of time-varying and directed graph networks remains an open question. Notably, it is not clear what is the effect of the network dynamics in the quality of the solution of specific problems such as the Wasserstein barycenter. Moreover, efficient communication strategies between nodes should be considered as well. The extension to the decentralized distributed setting of Sinkhorn-type algorithms [7] for regularized Wasserstein barycenter and other related algorithms, e.g., Wasserstein propagation [43], requires further work.

**Acknowledgments**

The work of A. Nedić and C.A. Uribe in Sect. 5 is supported by the National Science Foundation under grant no. CPS 15-44953. The research by P. Dvurechensky, D. Dvinskikh, and A. Gasnikov in Sect. 3 and Sect. 4 was funded by the Russian Science Foundation (project 18-71-10108).

## Footnotes

[1] Formally, the $\rho$-Wasserstein distance for $\rho \geq 1$ is $(\mathcal{W}_0(\mu, \nu))^{\frac{1}{\rho}}$ if $\mathcal{Y} = \mathcal{Z}$ and $c_i(y) = d^\rho(z_i, y)$, $d$ being a distance on $\mathcal{Y}$. For simplicity, we refer to (1) as regularized Wasserstein distance in a general situation since our algorithm does not rely on any specific choice of cost $c_i(y)$.

[2] For simplicity, we assume equal weights for each $\mathcal{W}_{\gamma,\mu_i}(p)$ and do not normalize the sum dividing by $m$. Our results can be directly generalized to the case of non-negative weights summing up to 1.

[3] Note that we can use also $\frac{1}{M_{k+1}}\sum_{r=1}^{M_{k+1}} p_i([\bar{\boldsymbol{\lambda}}_{k+1}]_i, Y_r^i)$ instead of $p_i([\bar{\boldsymbol{\lambda}}_{k+1}]_i, Y_1^i)$. This does not change the statement of Theorem 3, but reduces the variance of $\hat{\mathrm{p}}_N$ in practice. Thus, in the experiments, we use this estimator for the primal variable. Moreover, under mild assumptions, we can obtain high-probability analogue to inequalities (24).

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

# A   Proofs and Additional Numerical Results

In this appendix, we present the complete proofs of the Lemmas and Theorems stated in the main article. Moreover, we show additional experimental results. The contents of the Appendix are organized as follows:

- Subsection A.1, Subsection A.2, Subsection A.3, Subsection A.4, and Subsection A.5 present the complete proofs of the Lemmas and Theorems of the main paper.

- Subsection A.6 shows a graphic representation of the network topologies used for the experimental results, namely: complete graph, star graph, cycle graph and Erdős-Rényi random graph.

- Subsection A.7 shows, for various time instances, the local Wasserstein barycenter of 10 agents connected on an Erdős-Rényi random graph. Each agent holds a private Gaussian measure from which it can query samples. Different colors represent different agents. Time evolves with the number of iterations.

- Subsection A.8 shows, for various time instances, local Wasserstein barycenter of 10 agents connected on an Erdős-Rényi random graph. Each agent holds a private von Mises measure from which it can query samples. Different colors represent different agents. Time evolves with the number of iterations.

- Subsection A.9: shows, for various time instances, local Wasserstein barycenter of 100 agents connected on an Erdős-Rényi random graph. Each agent holds a private sample of the digit 2 from the MNIST dataset. We assume the normalize image as a probability distribution from which agents can sample from. Time evolves with the number of iterations.

- Subsection A.10: shows, for various time instances, local Wasserstein barycenter of 4 agents connected on an cycle graph. Each agent holds a private sample of an magnetic resonance image from the IXI dataset. We assume the normalize image as a probability distribution from which agents can sample from. Time evolves with the number of iterations.

- Attached videos

  - `Gauss_ex1.avi`: Example 1. The local Wasserstein barycenter of 10 agents connected on an Erdős-Rényi random graph. Each agent holds a private Gaussian measure from which it can query samples. Different colors represent different agents. Time evolves with the number of iterations.

  - `Gauss_ex2.avi`: Example 2. The local Wasserstein barycenter of 10 agents connected on an Erdős-Rényi random graph. Each agent holds a private Gaussian measure from which it can query samples. Different colors represent different agents. Time evolves with the number of iterations.

  - `MNIST_digit2.avi`: The local Wasserstein barycenter of 100 agents connected on an Erdős-Rényi random graph. Each agent holds a private sample of the digit 2 from the MNIST dataset. We assume the normalize image as a probability distribution from which agents can sample from. Time evolves with the number of iterations.

  - `MNIST_digit3.avi`: The local Wasserstein barycenter of 100 agents connected on an Erdős-Rényi random graph. Each agent holds a private sample of the digit 3 from the MNIST dataset. We assume the normalize image as a probability distribution from which agents can sample from. Time evolves with the number of iterations.

  - `von_mises_ex1.avi`: Example 1. The local Wasserstein barycenter of 10 agents connected on an Erdős-Rényi random graph. Each agent holds a private von Mises measure from which it can query samples. Different colors represent different agents. Time evolves with the number of iterations.

  - `von_mises_ex2.avi`: Example 2. The local Wasserstein barycenter of 10 agents connected on an Erdős-Rényi random graph. Each agent holds a private von Mises measure from which it can query samples. Different colors represent different agents. Time evolves with the number of iterations.

  - `ixi_mr.avi`: The local Wasserstein barycenter of 4 agents connected on an cycle graph. Each agent holds a private sample of an magnetic resonance image from the IXI dataset. We assume the normalize image as a probability distribution from which agents can sample from. Time evolves with the number of iterations.

## A.1 Proof of Lemma 1

Primal and dual optimal transport problem corresponding to the regularized Wasserstein distance can be written as follows

$$\mathcal{W}_{\gamma,\mu}(p) = \min_{\pi \in \Pi(\mu,\nu)} \left\{ \sum_{l=1}^{n} \int_{\mathcal{Y}} c_l(y)\pi_l(y)dy + \gamma \sum_{l=1}^{n} \int_{\mathcal{Y}} \pi_l(y)\log \pi_l(y)dy - \gamma \log \xi \right\}$$

$$= \max_{\bar{\lambda} \in \mathbb{R}^n, v \in \mathcal{C}(\mathcal{X})} \left\{ \sum_{l=1}^{n} [p]_l [\bar{\lambda}]_l + \int_{\mathcal{Y}} q(y)v(y)dy - \gamma \sum_{l=1}^{n} \int_{\mathcal{Y}} \exp\left( \frac{[\bar{\lambda}]_l - c_l(y) + v(y)}{\gamma} - 1 \right) dy \right\}$$

$$= \max_{\bar{\lambda} \in \mathbb{R}^n} \left\{ \langle p, \bar{\lambda} \rangle - \gamma \int_{\mathcal{Y}} \log\left( \frac{1}{q(y)} \sum_{l=1}^{n} \exp\left( \frac{[\bar{\lambda}]_l - c_l(y)}{\gamma} \right) \right) q(y)dy \right\},$$

where we used that $\xi$ is the uniform distribution on $\mathcal{Y} \times \mathcal{Z}$. By the definition of Fenchel-Legendre transform, using that $\mathcal{W}_{\gamma,\mu}(p) = (\mathcal{W}_{\gamma,\mu}(p))^{**}$, we get the first statement of the Lemma

$$\mathcal{W}_{\gamma,\mu}^{*}(\bar{\lambda}) = \gamma \int_{\mathcal{Y}} \log\left( \frac{1}{q(y)} \sum_{l=1}^{n} \exp\left( ([\bar{\lambda}]_l - c_l(y))/\gamma \right) \right) q(y)dy$$

$$= \mathbb{E}_{Y \sim \mu} \gamma \log\left( \frac{1}{q(Y)} \sum_{l=1}^{n} \exp\left( ([\bar{\lambda}]_l - c_l(Y))/\gamma \right) \right),$$

where $Y \sim \mu$ means that random variable $Y$ distributed according to measure $\mu$.

Differentiating, we obtain that the $l$-th component of the gradient of $\mathcal{W}_{\gamma,\mu}^{*}(\bar{\lambda})$ is

$$[\nabla \mathcal{W}_{\gamma,\mu}^{*}(\bar{\lambda})]_l = \int_{\mathcal{Y}} \frac{\exp(([\bar{\lambda}]_l - c_l(y))/\gamma)}{\sum_{\ell=1}^{n} \exp(([\bar{\lambda}]_\ell - c_\ell(y))/\gamma)} q(y)dy$$

$$= \mathbb{E}_{Y \sim \mu} \frac{\exp(([\bar{\lambda}]_l - c_l(Y))/\gamma)}{\sum_{\ell=1}^{n} \exp(([\bar{\lambda}]_\ell - c_\ell(Y))/\gamma)}, \quad l = 1, \dots, n.$$

To prove the Lipschitz continuity of this gradient, we calculate the diagonal elements of the Hessian

$$[\nabla^2 \mathcal{W}_{\gamma,\mu}^{*}(\bar{\lambda})]_{ll} = \frac{1}{\gamma} \int_{\mathcal{Y}} \frac{\exp(([\bar{\lambda}]_l - c_l(y))/\gamma) \sum_{\ell=1}^{n} \exp(([\bar{\lambda}]_\ell - c_\ell(y))/\gamma) - \exp^2(([\bar{\lambda}]_l - c_l(y))/\gamma)}{\left( \sum_{\ell=1}^{n} \exp(([\bar{\lambda}]_\ell - c_\ell(y))/\gamma) \right)^2} q(y)dy$$

and estimate its trace

$$\text{Tr}(\nabla^2 \mathcal{W}_{\gamma,\mu}^{*}(\bar{\lambda})) \le \frac{1}{\gamma} \int_{\mathcal{Y}} \frac{\sum_{l=1}^{n} \exp(([\bar{\lambda}]_l - c_l(y))/\gamma) \sum_{\ell=1}^{n} \exp(([\bar{\lambda}]_\ell - c_\ell(y))/\gamma)}{\left( \sum_{\ell=1}^{n} \exp(([\bar{\lambda}]_\ell - c_\ell(y))/\gamma) \right)^2} q(y)dy$$

$$= \frac{1}{\gamma} \int_{\mathcal{Y}} q(y)dy = \frac{1}{\gamma}.$$

This inequality proves that $\nabla \mathcal{W}_{\gamma,\mu}^{*}(\bar{\lambda})$ is $\frac{1}{\gamma}$- Lipschitz continuous with respect to the 2-norm.

## A.2 Proof of Theorem 1

Let us fix an arbitrary $\lambda \in \Lambda$. From the optimality condition in (13), we have

$$\langle \nabla V[\zeta_k](\zeta_{k+1}) + \alpha_{k+1}\nabla\Phi(\lambda_{k+1},\xi_{k+1}), \lambda - \zeta_{k+1} \rangle \ge 0. \tag{25}$$

Further,

$$\alpha_{k+1}\langle \nabla\Phi(\lambda_{k+1},\xi_{k+1}), \zeta_k - \lambda \rangle =$$

$$= \alpha_{k+1}\langle \nabla\Phi(\lambda_{k+1},\xi_{k+1}), \zeta_k - \zeta_{k+1} \rangle + \alpha_{k+1}\langle \nabla\Phi(\lambda_{k+1},\xi_{k+1}), \zeta_{k+1} - \lambda \rangle$$

$$\overset{(25)}{\le} \alpha_{k+1}\langle \nabla\Phi(\lambda_{k+1},\xi_{k+1}), \zeta_k - \zeta_{k+1} \rangle + \langle -\nabla V[\zeta_k](\zeta_{k+1}), \zeta_{k+1} - \lambda \rangle$$

$$= \alpha_{k+1}\langle \nabla\Phi(\lambda_{k+1},\xi_{k+1}), \zeta_k - \zeta_{k+1} \rangle + V[\zeta_k](\lambda) - V[\zeta_{k+1}](\lambda) - V[\zeta_k](\zeta_{k+1})$$

$$\overset{(10)}{\le} \alpha_{k+1}\langle \nabla\Phi(\lambda_{k+1},\xi_{k+1}), \zeta_k - \zeta_{k+1} \rangle + V[\zeta_k](\lambda) - V[\zeta_{k+1}](\lambda) - \frac{1}{2}\|\zeta_k - \zeta_{k+1}\|^2$$

$$\overset{(12),(14)}{=} C_{k+1}\langle \nabla\Phi(\lambda_{k+1},\xi_{k+1}), \lambda_{k+1} - \eta_{k+1} \rangle + V[\zeta_k](\lambda) - V[\zeta_{k+1}](\lambda) - \frac{C_{k+1}^2}{2\alpha_{k+1}^2}\|\lambda_{k+1} - \eta_{k+1}\|^2$$

$$\overset{(11)}{=} C_{k+1}\left( \langle \nabla\Phi(\lambda_{k+1},\xi_{k+1}), \lambda_{k+1} - \eta_{k+1} \rangle - \frac{2L}{2}\|\lambda_{k+1} - \eta_{k+1}\|^2 \right) + V[\zeta_k](\lambda) - V[\zeta_{k+1}](\lambda).$$

Add and subtract the term $C_{k+1}\langle\nabla\varphi(\lambda_{k+1}),\lambda_{k+1}-\eta_{k+1}\rangle$, then

$$\alpha_{k+1}\langle\nabla\Phi(\lambda_{k+1},\xi_{k+1}),\zeta_k-\lambda\rangle \le C_{k+1}\left(\langle\nabla\Phi(\lambda_{k+1},\xi_{k+1})-\nabla\varphi(\lambda_{k+1}),\lambda_{k+1}-\eta_{k+1}\rangle\right.$$
$$\left.-\frac{2L}{2}\|\lambda_{k+1}-\eta_{k+1}\|^2+\langle\nabla\varphi(\lambda_{k+1}),\lambda_{k+1}-\eta_{k+1}\rangle\right)+V[\zeta_k](\lambda)-V[\zeta_{k+1}](\lambda).\quad(26)$$

Using Fenchel inequality $\langle g,x\rangle\le\frac{1}{2\zeta}\|g\|_*^2+\frac{\zeta}{2}\|x\|^2$, we estimate

$$\langle\nabla\Phi(\lambda_{k+1},\xi_{k+1})-\nabla\varphi(\lambda_{k+1}),\lambda_{k+1}-\eta_{k+1}\rangle \le$$
$$\le\frac{1}{2L}\|\nabla\Phi(\lambda_{k+1},\xi_{k+1})-\nabla\varphi(\lambda_{k+1})\|_*^2+\frac{L}{2}\|\lambda_{k+1}-\eta_{k+1}\|^2.$$

Therefore, we can rewrite (26) as

$$\alpha_{k+1}\langle\nabla\Phi(\lambda_{k+1},\xi_{k+1}),\zeta_k-\lambda\rangle \le$$
$$\le C_{k+1}\left(\frac{1}{2L}\|\nabla\Phi(\lambda_{k+1},\xi_{k+1})-\nabla\varphi(\lambda_{k+1})\|_*^2\right.$$
$$\left.+\langle\nabla\varphi(\lambda_{k+1}),\lambda_{k+1}-\eta_{k+1}\rangle-\frac{L}{2}\|\lambda_{k+1}-\eta_{k+1}\|^2\right)+V[\zeta_k](\lambda)-V[\zeta_{k+1}](\lambda)$$
$$=C_{k+1}\left(\langle\nabla\varphi(\lambda_{k+1}),\lambda_{k+1}-\eta_{k+1}\rangle-\frac{L}{2}\|\lambda_{k+1}-\eta_{k+1}\|^2\right)+V[\zeta_k](\lambda)-V[\zeta_{k+1}](\lambda)$$
$$+\frac{C_{k+1}}{2L}\|\nabla\Phi(\lambda_{k+1},\xi_{k+1})-\nabla\varphi(\lambda_{k+1})\|_*^2$$
$$\overset{(7)}{\le}C_{k+1}\left(\varphi(\lambda_{k+1})-\varphi(\eta_{k+1})\right)+V[\zeta_k](\lambda)-V[\zeta_{k+1}](\lambda)$$
$$+\frac{C_{k+1}}{2L}\|\nabla\Phi(\lambda_{k+1},\xi_{k+1})-\nabla\varphi(\lambda_{k+1})\|_*^2.\quad(27)$$

Similarly, adding and substracting the term $\langle\nabla\varphi(\lambda_{k+1}),\eta_k-\lambda_{k+1}\rangle$, we have

$$\langle\nabla\Phi(\lambda_{k+1},\xi_{k+1}),\eta_k-\lambda_{k+1}\rangle \le$$
$$\le\langle\nabla\varphi(\lambda_{k+1}),\eta_k-\lambda_{k+1}\rangle+\langle\nabla\Phi(\lambda_{k+1},\xi_{k+1})-\nabla\varphi(\lambda_{k+1}),\eta_k-\lambda_{k+1}\rangle$$
$$\overset{\text{conv-ty}}{\le}\varphi(\eta_k)-\varphi(\lambda_{k+1})+\langle\nabla\Phi(\lambda_{k+1},\xi_{k+1})-\nabla\varphi(\lambda_{k+1}),\eta_k-\lambda_{k+1}\rangle.\quad(28)$$

Finally,

$$\alpha_{k+1}\langle\nabla\Phi(\lambda_{k+1},\xi_{k+1}),\lambda_{k+1}-\lambda\rangle =$$
$$=\alpha_{k+1}\langle\nabla\Phi(\lambda_{k+1},\xi_{k+1}),\lambda_{k+1}-\zeta_k\rangle+\alpha_{k+1}\langle\nabla\Phi(\lambda_{k+1},\xi_{k+1}),\zeta_k-\lambda)\rangle$$
$$\overset{(11),(12)}{=}C_k\langle\nabla\Phi(\lambda_{k+1},\xi_{k+1}),\eta_k-\lambda_{k+1}\rangle+\alpha_{k+1}\langle\nabla\Phi(\lambda_{k+1},\xi_{k+1}),\zeta_k-\lambda\rangle$$
$$\overset{(27),(28)}{\le}C_k\left(\varphi(\eta_k)-\varphi(\lambda_{k+1})+\langle\nabla\Phi(\lambda_{k+1},\xi_{k+1})-\nabla\varphi(\lambda_{k+1}),\eta_k-\lambda_{k+1}\rangle\right)$$
$$+C_{k+1}\left(\varphi(\lambda_{k+1})-\varphi(\eta_{k+1})\right)+V[\zeta_k](\lambda)-V[\zeta_{k+1}](\lambda)$$
$$+\frac{C_{k+1}}{2L}\|\nabla\Phi(\lambda_{k+1},\xi_{k+1})-\nabla\varphi(\lambda_{k+1})\|_*^2$$
$$\overset{(11)}{=}\alpha_{k+1}\varphi(\lambda_{k+1})+C_k\varphi(\eta_k)-C_{k+1}\varphi(\eta_{k+1})+V[\zeta_k](\lambda)-V[\zeta_{k+1}](\lambda)$$
$$+C_{k+1}\langle\nabla\Phi(\lambda_{k+1},\xi_{k+1})-\nabla\varphi(\lambda_{k+1}),\eta_k-\lambda_{k+1}\rangle$$
$$+\frac{C_{k+1}}{2L}\|\nabla\Phi(\lambda_{k+1},\xi_{k+1})-\nabla\varphi(\lambda_{k+1})\|_*^2$$

Rearranging terms, we obtain

$$C_{k+1}\varphi(\eta_{k+1})-C_k\varphi(\eta_k) \le$$
$$\le\alpha_{k+1}\left(\varphi(\lambda_{k+1})+\langle\nabla\Phi(\lambda_{k+1},\xi_{k+1}),\lambda-\lambda_{k+1}\rangle\right)+V[\zeta_k](\lambda)-V[\zeta_{k+1}](\lambda)$$
$$+C_{k+1}\langle\nabla\Phi(\lambda_{k+1},\xi_{k+1})-\nabla\varphi(\lambda_{k+1}),\eta_k-\lambda_{k+1}\rangle$$
$$+\frac{C_{k+1}}{2L}\|\nabla\Phi(\lambda_{k+1},\xi_{k+1})-\nabla\varphi(\lambda_{k+1})\|_*^2.$$

Summing these inequalities for $k = 0, \dots, N - 1$, we get

$$C_N \varphi(\eta_N) - C_0 \varphi(\eta_0) \leq \sum_{k=0}^{N-1} \alpha_{k+1} \left( \varphi(\lambda_{k+1}) + \langle \nabla \Phi(\lambda_{k+1}, \xi_{k+1}), \lambda - \lambda_{k+1} \rangle \right)$$

$$+ V[\zeta_0](\lambda) - V[\zeta_N](\lambda) + \sum_{k=0}^{N-1} C_{k+1} \langle \nabla \Phi(\lambda_{k+1}, \xi_{k+1}) - \nabla \varphi(\lambda_{k+1}), \eta_k - \lambda_{k+1} \rangle +$$

$$+ \sum_{k=0}^{N-1} \frac{C_{k+1}}{2L} \| \nabla \Phi(\lambda_{k+1}, \xi_{k+1}) - \nabla \varphi(\lambda_{k+1}) \|_*^2.$$

Since $C_0 = \alpha_0 = 0$ and $V[\zeta_k](\lambda) \geq 0$, we end up with

$$C_N \varphi(\eta_N) \leq \sum_{k=0}^{N} \alpha_k \left( \varphi(\lambda_k) + \langle \nabla \Phi(\lambda_k, \xi_k), \lambda - \lambda_k \rangle \right) + V[\zeta_0](\lambda)$$

$$+ \sum_{k=0}^{N-1} C_{k+1} \langle \nabla \Phi(\lambda_{k+1}, \xi_{k+1}) - \nabla \varphi(\lambda_{k+1}), \eta_k - \lambda_{k+1} \rangle + \sum_{k=0}^{N} \frac{C_k}{2L} \| \nabla \Phi(\lambda_k, \xi_k) - \nabla \varphi(\lambda_k) \|_*^2.$$

Since $\lambda \in \Lambda$ was chosen arbitrarily, we take the minimum in $\lambda$ in the right hand side of this inequality and obtain the statement of the Theorem.

### A.3   Proof of Theorem 2

Let us introduce a set $\Lambda_R := \{ \lambda \in H^* : \|\lambda\|_2 \leq 2R \}$. Then, from (15) since $\zeta_0 = 0$ and $V[\zeta](\lambda) = \frac{1}{2} \|\lambda - \zeta\|_2^2$, we have

$$C_N \varphi(\eta_N) \leq \min_{\lambda \in \Lambda} \left\{ \sum_{k=0}^{N} \alpha_k \left( \varphi(\lambda_k) + \langle \nabla \Phi(\lambda_k, \xi_k), \lambda - \lambda_k \rangle \right) + \frac{1}{2} \|\lambda\|_2^2 \right\}$$

$$+ \sum_{k=0}^{N-1} C_{k+1} \langle \nabla \Phi(\lambda_{k+1}, \xi_{k+1}) - \nabla \varphi(\lambda_{k+1}), \eta_k - \lambda_{k+1} \rangle + \sum_{k=0}^{N} \frac{C_k}{2L} \| \nabla \Phi(\lambda_k, \xi_k) - \nabla \varphi(\lambda_k) \|_2^2.$$

$$\leq \min_{\lambda \in \Lambda_R} \left\{ \sum_{k=0}^{N} \alpha_k \left( \varphi(\lambda_k) + \langle \nabla \Phi(\lambda_k, \xi_k), \lambda - \lambda_k \rangle \right) + \frac{1}{2} \|\lambda\|_2^2 \right\}$$

$$+ \sum_{k=0}^{N-1} C_{k+1} \langle \nabla \Phi(\lambda_{k+1}, \xi_{k+1}) - \nabla \varphi(\lambda_{k+1}), \eta_k - \lambda_{k+1} \rangle + \sum_{k=0}^{N} \frac{C_k}{2L} \| \nabla \Phi(\lambda_k, \xi_k) - \nabla \varphi(\lambda_k) \|_2^2$$

$$\leq \min_{\lambda \in \Lambda_R} \left\{ \sum_{k=0}^{N} \alpha_k \left( \varphi(\lambda_k) + \langle \nabla \Phi(\lambda_k, \xi_k), \lambda - \lambda_k \rangle \right) \right\} + 2R^2$$

$$+ \sum_{k=0}^{N-1} C_{k+1} \langle \nabla \Phi(\lambda_{k+1}, \xi_{k+1}) - \nabla \varphi(\lambda_{k+1}), \eta_k - \lambda_{k+1} \rangle + \sum_{k=0}^{N} \frac{C_k}{2L} \| \nabla \Phi(\lambda_k, \xi_k) - \nabla \varphi(\lambda_k) \|_2^2. \quad (29)$$

Our next goal is to take the expectation from the both sides of this inequality with respect to the seqence $\xi_0, \dots, \xi_N$. To do so, we iteratively, for each $j$ from $N$ to $0$ fix the history $\xi_0, \dots, \xi_{j-1}$ and take the expectation w.r.t $\xi_j$.

Since $\mathbb{E}_{\xi_{k+1}} [\nabla \Phi(\lambda_{k+1}, \xi_{k+1}) | \xi_1, \dots, \xi_k] = \nabla \varphi(\lambda_{k+1})$, $\lambda_{k+1}$ and $\eta_k$ are deterministic functions of $(\xi_1, \dots, \xi_k)$, we have $\mathbb{E}_{\xi_1, \dots \xi_k} \langle \nabla \Phi(\lambda_{k+1}, \xi_{k+1}) - \nabla \varphi(\lambda_{k+1}), \eta_k - \lambda_{k+1} \rangle = 0$. By the Theorem assumption

$$\mathbb{E}_{\xi_k} \left[ \| \nabla \Phi(\lambda_k, \xi_k) - \nabla \varphi(\lambda_k) \|_2^2 | \xi_1, \dots, \xi_{k-1} \right] \leq \frac{\varepsilon L \alpha_k}{C_k}.$$

Thus, after taking the full expectation $\mathbb{E}$, the last three terms in the r.h.s. of (29) satisfy

$$\mathbb{E} \left[ 2R^2 + \sum_{k=0}^{N-1} C_{k+1} \langle \nabla \Phi(\lambda_{k+1}, \xi_{k+1}) - \nabla \varphi(\lambda_{k+1}), \eta_k - \lambda_{k+1} \rangle + \sum_{k=0}^{N} \frac{C_k}{2L} \| \nabla \Phi(\lambda_k, \xi_k) - \nabla \varphi(\lambda_k) \|_2^2 \right]$$

$$\leq 2R^2 + \frac{C_N \varepsilon}{2}, \quad (30)$$

where we used that $C_N = \sum_{k=0}^{N} \alpha_k$.

Let us now estimate the expectation of the first term in the r.h.s. of (29). By the definition of $F(x, \xi)$ and $F^*(-A^T \lambda, \xi)$ in subsection 3.1, we have

$$F^*(-A^T \lambda_k, \xi_k) + \langle A \nabla F^*(-A^T \lambda_k, \xi_k), \lambda_k \rangle = \langle -A^T \lambda_k, x(-A^T \lambda_k, \xi_k) \rangle$$
$$- F(x(-A^T \lambda_k, \xi_k), \xi_k) + \langle Ax(-A^T \lambda_k, \xi_k), \lambda_k \rangle$$
$$= -F(x(-A^T \lambda_k, \xi_k), \xi_k). \tag{31}$$

On the other hand, by Fenchel duality,

$$\mathbb{E}_{\xi_k} F(x(-A^T \lambda_k, \xi_k), \xi_k) = \mathbb{E}_{\xi_k} \max_{\tilde{\lambda}} \{ \langle x(-A^T \lambda_k, \xi_k), \tilde{\lambda} \rangle - F^*(\tilde{\lambda}, \xi) \}$$
$$\geq \max_{\tilde{\lambda}} \{ \langle \mathbb{E}_{\xi_k} x(-A^T \lambda_k, \xi_k), \tilde{\lambda} \rangle - \mathbb{E}_{\xi_k} F^*(\tilde{\lambda}, \xi) \} = f \left( \mathbb{E}_{\xi_k} x(-A^T \lambda_k, \xi_k) \right). \tag{32}$$

Hence,

$$\mathbb{E}_{\xi_k} (F^*(-A^T \lambda_k, \xi_k) + \langle A \nabla F^*(-A^T \lambda_k, \xi_k), \lambda_k \rangle) \leq -f(\mathbb{E}_{\xi_k} x(-A^T \lambda_k, \xi_k))$$

Using this inequality, (8) and that $\nabla \Phi(\lambda_k, \xi_k) = b - A \nabla F^*(-A^T \lambda_k, \xi_k) = b - Ax(-A^T \lambda_k, \xi_k)$, we obtain

$$\mathbb{E}_{\xi_k} (\varphi(\lambda_k) + \langle \nabla \Phi(\lambda_k, \xi_k), \lambda - \lambda_k \rangle) = \langle b, \lambda_k \rangle + \mathbb{E}_{\xi_k} F^*(-A^T \lambda_k, \xi_k) + \mathbb{E}_{\xi_k} \langle b - A \nabla F^*(-A^T \lambda_k, \xi_k), \lambda - \lambda_k \rangle$$
$$= \mathbb{E}_{\xi_k} (F^*(-A^T \lambda_k, \xi_k) + \langle A \nabla F^*(-A^T \lambda_k, \xi_k), \lambda_k \rangle)$$
$$+ \mathbb{E}_{\xi_k} \langle b - Ax(-A^T \lambda_k, \xi_k), \lambda \rangle$$
$$\leq -f(\mathbb{E}_{\xi_k} x(-A^T \lambda_k, \xi_k)) + \langle b - A \mathbb{E}_{\xi_k} x(-A^T \lambda_k, \xi_k), \lambda \rangle. \tag{33}$$

Taking the full expectation from the first term in the r.h.s. of (29) and iteratively applying (33), we obtain

$$\mathbb{E} \min_{\lambda \in \Lambda_R} \left\{ \sum_{k=0}^{N} \alpha_k (\varphi(\lambda_k) + \langle \nabla \Phi(\lambda_k, \xi_k), \lambda - \lambda_k \rangle) \right\} \leq \min_{\lambda \in \Lambda_R} \left\{ \mathbb{E} \sum_{k=0}^{N} \alpha_k (\varphi(\lambda_k) + \langle \nabla \Phi(\lambda_k, \xi_k), \lambda - \lambda_k \rangle) \right\}$$

$$\leq \min_{\lambda \in \Lambda_R} \left\{ \sum_{k=0}^{N} \alpha_k (-f(\mathbb{E}x(-A^T \lambda_k, \xi_k)) + \langle b - A \mathbb{E}x(-A^T \lambda_k, \xi_k), \lambda \rangle) \right\}$$
$$\leq C_N \min_{\lambda \in \Lambda_R} \{ -f(\mathbb{E}\hat{x}_N) + \langle b - A \mathbb{E}\hat{x}_N, \lambda \rangle \} \leq -C_N f(\mathbb{E}\hat{x}_N) + C_N \min_{\lambda \in \Lambda_R} \langle b - A \mathbb{E}\hat{x}_N, \lambda \rangle$$
$$= -C_N f(\mathbb{E}\hat{x}_N) - 2C_N R \| b - A \mathbb{E}\hat{x}_N \|_2, \tag{34}$$

where we also used the convexity of $f$, equality $\sum_{k=0}^{N} \alpha_k = C_N$, and definitions of $\hat{x}_N$ and $\Lambda_R$.

Taking the expectation in (29) and combining it with (30) and (34), we obtain

$$\mathbb{E}\varphi(\eta_N) + f(\mathbb{E}\hat{x}_N) \leq -2R \| A \mathbb{E}\hat{x}_N - b \|_2 + \frac{2R^2}{C_N} + \frac{\varepsilon}{2}. \tag{35}$$

Hence, by weak duality $-f(x^*) \leq \varphi(\eta^*)$,

$$f(\mathbb{E}\hat{x}_N) - f(x^*) \leq f(\mathbb{E}\hat{x}_N) + \varphi(\eta^*) \leq f(\mathbb{E}\hat{x}_N) + \mathbb{E}\varphi(\eta_N) \leq \frac{2R^2}{C_N} + \frac{\varepsilon}{2}. \tag{36}$$

Since $\lambda^*$ is an optimal solution of Problem (D), we have, for any $x \in Q$, $f(x^*) \leq f(x) + \langle \lambda^*, Ax - b \rangle$. Then using assumption $\| \lambda^* \|_2 \leq R$ and choosing $x = \mathbb{E}\hat{x}_N$, we get

$$f(\mathbb{E}\hat{x}_N) \geq f(x^*) - R \| A \mathbb{E}\hat{x}_N - b \|_2 \tag{37}$$

Using this and weak duality $-f(x^*) \leq \varphi(\eta^*)$ and taking the expectation, we obtain

$$\mathbb{E}\varphi(\eta_N) + f(\mathbb{E}\hat{x}_N) \geq \varphi(\eta^*) + f(\mathbb{E}\hat{x}_N) \geq -f(x^*) + f(\mathbb{E}\hat{x}_N) \overset{(37)}{\geq} -R \| A \mathbb{E}\hat{x}_N - b \|_2$$

Using this and (35), we get

$$\| A \mathbb{E}\hat{x}_N - b \|_2 \leq \frac{2R}{C_N} + \frac{\varepsilon}{2R} \tag{38}$$

It remains to estimate the growth of coefficients $C_N$. So, we prove by induction that the coefficients $C_k$ generated by Algorithm 4 satisfy the following condition

$$C_k \geq \frac{(k+1)^2}{8L}. \tag{39}$$

Since $C_0 = 0$ for $k = 1$ $C_1 \overset{(16)}{=} \frac{1}{2L}$ and (39) holds. Let us now assume that (39) holds for some $k \geq 1$ and prove that it holds for $k + 1$. By (11), $\alpha_{k+1}$ is the largest root of the equation $2L\alpha_{k+1}^2 - \alpha_{k+1} - C_k = 0$. Thus,

$$\alpha_{k+1} = \frac{1 + \sqrt{1 + 8LC_k}}{4L} = \frac{1}{4L} + \sqrt{\frac{1}{8L^2} + \frac{C_k}{2L}} \geq \frac{1}{4L} + \sqrt{\frac{C_k}{2L}} \geq \frac{1}{4L} + \frac{1}{\sqrt{2L}} \frac{k+1}{2\sqrt{2L}} = \frac{k+2}{4L}. \tag{40}$$

Using the induction assumption, (11), (39) and (40), we obtain that (39) holds for $k + 1$

$$C_{k+1} = C_k + \alpha_{k+1} \geq \frac{(k+1)^2}{8L} + \frac{k+2}{4L} \geq \frac{(k+2)^2}{8L}.$$

Combining (36), (38), and (39), we finish our proof.

## A.4 Proof of Lemma 2

First, let us estimate the Lipschitz constant of $\nabla \mathcal{W}_\gamma^*(\boldsymbol{\lambda})$

$$
\begin{aligned}
\|\nabla \mathcal{W}_\gamma^*(\boldsymbol{\lambda}_1) - \nabla \mathcal{W}_\gamma^*(\boldsymbol{\lambda}_2)\|_2^2 &\overset{(6)}{=} \left\| \sqrt{W} \begin{pmatrix} \nabla \mathcal{W}_{\gamma,\mu_1}^*([\bar{\boldsymbol{\lambda}}_1]_1) \\ ... \\ \nabla \mathcal{W}_{\gamma,\mu_m}^*([\bar{\boldsymbol{\lambda}}_1]_m) \end{pmatrix} - \sqrt{W} \begin{pmatrix} \nabla \mathcal{W}_{\gamma,\mu_1}^*([\bar{\boldsymbol{\lambda}}_2]_1) \\ ... \\ \nabla \mathcal{W}_{\gamma,\mu_m}^*([\bar{\boldsymbol{\lambda}}_2]_m) \end{pmatrix} \right\|_2^2 \\
&\leq (\lambda_{\max}(\sqrt{W}))^2 \left\| \begin{matrix} \nabla \mathcal{W}_{\gamma,\mu_1}^*([\bar{\boldsymbol{\lambda}}_1]_1) - \nabla \mathcal{W}_{\gamma,\mu_1}^*([\bar{\boldsymbol{\lambda}}_2]_1) \\ ... \\ \nabla \mathcal{W}_{\gamma,\mu_m}^*([\bar{\boldsymbol{\lambda}}_1]_m) - \nabla \mathcal{W}_{\gamma,\mu_m}^*([\bar{\boldsymbol{\lambda}}_2]_m) \end{matrix} \right\|_2^2 \\
&= (\lambda_{\max}(\sqrt{W}))^2 \sum_{i=1}^m \left\| \nabla \mathcal{W}_{\gamma,\mu_i}^*([\bar{\boldsymbol{\lambda}}_1]_i) - \nabla \mathcal{W}_{\gamma,\mu_i}^*([\bar{\boldsymbol{\lambda}}_2]_i) \right\|_2^2 \\
&\leq (\lambda_{\max}(\sqrt{W}))^2 \sum_{i=1}^m \frac{1}{\gamma^2} \left\| [\bar{\boldsymbol{\lambda}}_1]_i - [\bar{\boldsymbol{\lambda}}_2]_i \right\|_2^2 \\
&= \frac{(\lambda_{\max}(\sqrt{W}))^2}{\gamma^2} \sum_{i=1}^m \left\| [\sqrt{W}(\boldsymbol{\lambda}_1 - \boldsymbol{\lambda}_2)]_i \right\|_2^2 \\
&= \frac{(\lambda_{\max}(\sqrt{W}))^2}{\gamma^2} \left\| \sqrt{W}(\boldsymbol{\lambda}_1 - \boldsymbol{\lambda}_2) \right\|_2^2 \\
&\leq \frac{(\lambda_{\max}(\sqrt{W}))^4}{\gamma^2} \left\| \boldsymbol{\lambda}_1 - \boldsymbol{\lambda}_2 \right\|_2^2,
\end{aligned}
$$

where we used notation $\bar{\boldsymbol{\lambda}} = \sqrt{W}\boldsymbol{\lambda}$, the definition of matrix $\sqrt{W}$, $1/\gamma$-Lipschitz continuity of $\nabla \mathcal{W}_{\gamma,\mu_i}^*(\bar{\lambda}_i)$ for all $i = 1, \ldots, m$. Since $(\lambda_{\max}(\sqrt{W}))^4 = (\lambda_{\max}(W))^2$, we obtain that the dual function $\mathcal{W}_\gamma^*(\boldsymbol{\lambda})$ has $\lambda_{\max}(W)/\gamma$-Lipschitz continuous gradient.

By Lemma 1, vectors $p_j(\bar{\lambda}_j, Y_r^j)$, $j = 1, ..., m$, $r = 1, ..., M$ defined in (23) satisfy $\mathbb{E}_{Y_r^j} p_j(\bar{\lambda}_j, Y_r^j) = \nabla \mathcal{W}_{\gamma,\mu_j}^*(\bar{\lambda}_j)$. Thus, by (6), (21), (22) we have $\mathbb{E}\widetilde{\nabla}\mathcal{W}_\gamma^*(\boldsymbol{\lambda}) = \nabla \mathcal{W}_\gamma^*(\boldsymbol{\lambda})$.

Further, for $j = 1, ..., m$, we estimate the variance of $p_j(\bar{\lambda}_j, Y^j)$

$$
\begin{aligned}
\mathbb{E}_{Y^j \sim \mu_j} \|p_j(\bar{\lambda}_j, Y^j) - \nabla \mathcal{W}_{\gamma,\mu_j}^*(\bar{\lambda}_j)\|_2^2 &= \mathbb{E}_{Y^j \sim \mu_j} \sum_{l=1}^n \left( \frac{\exp\left(([\bar{\lambda}_j]_l - c_l(Y))/\gamma\right)}{\sum_{\ell=1}^n \exp\left(([\bar{\lambda}_j]_\ell - c_\ell(Y))/\gamma\right)} - [\nabla \mathcal{W}_{\gamma,\mu_j}^*(\bar{\lambda}_j)]_l \right)^2 \\
&= \sum_{l=1}^n \mathbb{E}_{Y^j \sim \mu_j} \frac{\exp^2\left(([\bar{\lambda}_j]_l - c_l(Y))/\gamma\right)}{\left(\sum_{\ell=1}^n \exp\left(([\bar{\lambda}_j]_\ell - c_\ell(Y))/\gamma\right)\right)^2} - \sum_{l=1}^n [\nabla \mathcal{W}_{\gamma,\mu_j}^*(\bar{\lambda}_j)]_l^2 \\
&\leq \sum_{l=1}^n \int_{\mathcal{Y}} \frac{\exp^2(([\bar{\lambda}_j]_l - c_l(y))/\gamma) q(y)}{\left(\sum_{\ell=1}^n \exp(([\bar{\lambda}]_j]_\ell - c_\ell(y))/\gamma)\right)^2} q(y) dy \\
&= \int_{\mathcal{Y}} \frac{\sum_{l=1}^n \exp^2(([\bar{\lambda}_j]_l - c(y,z_i))/\gamma)}{\left(\sum_{\ell=1}^n \exp(([\bar{\lambda}]_j]_\ell - c_\ell(y))/\gamma)\right)^2} q(y) dy \leq \int_{\mathcal{Y}} q(y) dy = 1.
\end{aligned}
$$

Hence, by (22), for $j = 1, ..., m$, we have

$$\mathbb{E}_{Y_r^j \sim \mu_j, r=1,...,M} \|\widetilde{\nabla}\mathcal{W}_{\gamma,\mu_j}^*(\bar{\lambda}_j) - \nabla \mathcal{W}_{\gamma,\mu_j}^*(\bar{\lambda}_j)\|_2^2 \leq \frac{1}{M}. \tag{41}$$

By the same arguments as above for the estimate of the Lipschitz constant for $\nabla \mathcal{W}_\gamma^*(\boldsymbol{\lambda})$, we estimate the variance of $\widetilde{\nabla} \mathcal{W}_\gamma^*(\boldsymbol{\lambda})$. Denoting $\mathbb{E} = \mathbb{E}_{Y_r^j \sim \mu_j, j=1,\ldots,m, r=1,\ldots,M}$, we have

$$
\mathbb{E} \|\widetilde{\nabla} \mathcal{W}_\gamma^*(\boldsymbol{\lambda}) - \nabla \mathcal{W}_\gamma^*(\boldsymbol{\lambda})\|_2^2 \overset{(6),(21)}{=} \mathbb{E} \left\| \sqrt{W} \begin{pmatrix} \widetilde{\nabla} \mathcal{W}_{\gamma,\mu_1}^*([\bar{\boldsymbol{\lambda}}]_1) \\ \ldots \\ \widetilde{\nabla} \mathcal{W}_{\gamma,\mu_m}^*([\bar{\boldsymbol{\lambda}}]_m) \end{pmatrix} - \sqrt{W} \begin{pmatrix} \nabla \mathcal{W}_{\gamma,\mu_1}^*([\bar{\boldsymbol{\lambda}}]_1) \\ \ldots \\ \nabla \mathcal{W}_{\gamma,\mu_m}^*([\bar{\boldsymbol{\lambda}}]_m) \end{pmatrix} \right\|_2^2
$$

$$
\leq (\lambda_{\max}(\sqrt{W}))^2 \mathbb{E} \left\| \begin{matrix} \widetilde{\nabla} \mathcal{W}_{\gamma,\mu_1}^*([\bar{\boldsymbol{\lambda}}]_1) - \nabla \mathcal{W}_{\gamma,\mu_1}^*([\bar{\boldsymbol{\lambda}}]_1) \\ \ldots \\ \widetilde{\nabla} \mathcal{W}_{\gamma,\mu_m}^*([\bar{\boldsymbol{\lambda}}]_m) - \nabla \mathcal{W}_{\gamma,\mu_m}^*([\bar{\boldsymbol{\lambda}}]_m) \end{matrix} \right\|_2^2
$$

$$
= (\lambda_{\max}(\sqrt{W}))^2 \mathbb{E} \sum_{i=1}^m \left\| \widetilde{\nabla} \mathcal{W}_{\gamma,\mu_i}^*([\bar{\boldsymbol{\lambda}}]_i) - \nabla \mathcal{W}_{\gamma,\mu_i}^*([\bar{\boldsymbol{\lambda}}]_i) \right\|_2^2
$$

$$
\overset{(41)}{\leq} \frac{(\lambda_{\max}(\sqrt{W}))^2 m}{M} = \frac{\lambda_{\max}(W) m}{M},
$$

which finishes the proof of the Lemma.

## A.5    Proof of Theorem 3

Combining Lemma 2 and Theorem 2 for our particular case of primal-dual pair of problems (4)-(5) with $A = \sqrt{W}$, $b = 0$, $L = \lambda_{\max}(W)/\gamma$, since $N = \sqrt{32\lambda_{\max}(W)R^2/(\varepsilon\gamma)}$, we obtain the first statement of the theorem.

Let us now estimate the overall complexity of Algorithm 4. For each agent $i$, the complexity of each iteration is dominated by the complexity of calculation of stochastic approximation $\widetilde{\nabla} \mathcal{W}_{\gamma,\mu_i}^*([\bar{\boldsymbol{\lambda}}_k]_i)$ for the gradient. This complexity is $O(mnM_k)$. Thus, to get the overall complexity, we need to estimate $\sum_{k=1}^N M_k$

$$
\sum_{k=1}^N M_k = \sum_{k=1}^N \max\left\{ 1, \left\lceil \frac{m\gamma C_k}{\alpha_k \varepsilon} \right\rceil \right\} \overset{(16)}{\leq} \max\left\{ N, \left\lceil \frac{2\lambda_{\max}(W)m}{\varepsilon} \sum_{k=1}^N \alpha_k \right\rceil \right\}
$$

$$
= \max\left\{ N, \left\lceil \frac{2\lambda_{\max}(W)m}{\varepsilon} C_N \right\rceil \right\}
$$

where we used that $\sum_{k=1}^N \alpha_k = C_N$. From (38) and definition of $N$ it follows that

$$
\frac{2R}{C_N} \leq \frac{\varepsilon}{2R} \quad \text{and} \quad \frac{2R}{C_{N-1}} \geq \frac{\varepsilon}{2R}.
$$

Then

$$
C_{N-1} \leq 4R^2/\varepsilon \tag{42}
$$

From (40)

$$
\alpha_N = \frac{1}{4L} + \sqrt{\frac{1}{8L^2} + \frac{C_{N-1}}{2L}} \leq \frac{1}{2L} + \sqrt{\frac{C_{N-1}}{2L}} \overset{(16)}{=} \frac{1}{2L} + \alpha_{N-1} \tag{43}
$$

On the other hand, from (40) it follows that

$$
\alpha_{N-1} = \frac{1}{4L} + \sqrt{\frac{1}{8L^2} + \frac{C_{N-2}}{2L}} \geq \frac{1}{4L} \tag{44}
$$

Hence, from (43) and (44) we have

$$
\alpha_N \leq 2\alpha_{N-1} + \alpha_{N-1} = 3\alpha_{N-1} \overset{(16)}{\leq} 3C_{N-1}
$$

Since this inequality and (16) we obtain $C_N \leq 4C_{N-1}$. Then using (42) we have

$$
\sum_{k=1}^N M_k \leq \max\left\{ \sqrt{\frac{32\lambda_{\max}(W)R^2}{\varepsilon\gamma}}, \frac{32\lambda_{\max}(W)mR^2}{\varepsilon^2} \right\}, \tag{45}
$$

where in last equality we used $N = \sqrt{32\lambda_{\max}(W)R^2/(\varepsilon\gamma)}$. To obtain the total complexity, we multiply the above estimate for $\sum_{k=1}^N M_k$ by $mn$.

## A.6 Visualization of the Network Topologies used in Simulations

(a) Star Graph

(b) Cycle Graph

(c) Erdős-Rényi random graph

(d) Complete Graph

Figure 5: Example of Network Topologies.

## A.7 Additional Example on Gaussian Distributions

Figure 6: Local Wasserstein barycenter of 10 agents connected on an Erdős-Rényi random graph. Each agent holds a private Gaussian measure from which it can query samples. Different colors represent different agents.

## A.8 Additional Example on von Mises Distributions

Figure 7: Local Wasserstein barycenter of 10 agents connected on an Erdős-Rényi random graph. Each agent holds a private von Mises measure from which it can query samples. Different colors represent different agents.

## A.9 Additional Information for the MNIST Dataset

Figure 8: Local Wasserstein barycenter of $100$ agents connected on an Erdős-Rényi random graph. Each agent holds a private sample of the digit 2 from the MNIST dataset. We assume the normalize image as a probability distribution from which agents can sample from.

## A.10 Additional Information for the IXI Dataset

Figure 9: The samples from the IXI dataset held by four agents.

Figure 10: Local Wasserstein barycenter of 4 agents connected on a cycle graph. Each agent holds a private sample of an magnetic resonance image from the IXI dataset. We assume the normalize image as a probability distribution from which agents can sample from. Time evolves with the number of iterations.