[Reviews · NeurIPS 2018]

Reviewer 1



This paper presents a distributed algorithm for computing Wasserstein barycenters. The basic setup is that each agent in the decentralized system has access to one probability distribution; similar to “gossip” based optimization techniques in the classical case (e.g. popularized via ADMM-style techniques) here the distributed algorithm exchanges samples from local measures along edges of a graph. It seems this paper missed the closest related work, “Stochastic Wasserstein Barycenters” (Claici et al., ArXiv/ICML), which proposes a nonconvex semidiscrete barycenter optimization algorithm. Certainly any final version of this paper needs to compare to that work carefully. It may also be worth noting that the Wasserstein propagation algorithm in “Convolutional Wasserstein Distances: Efficient Optimal Transportation on Geometric Domains” (2015) could be implemented easily on a network in a similar fashion to what is proposed in this paper; see their Algorithm 4. Like lots of previous work in OT, this technique uses entropic regularization to make transport tractable; they solve the smoothed dual. The basic model for translating this problem to a network problem is similar to that in many other distributed optimization algorithm; you duplicate the unknown barycenter variable per vertex and add equality conditions along the edges. In general, I found the idea in this paper to be compelling, and I appreciate the careful theoretical and experimental analysis. I’m not sure the paper actually makes a strong case for network-based optimization (I’m guessing the experiments here were largely on a single machine than some huge network of computers), and I’d love to see truly large-scale experiments that better justify ML applications----but this can be a great topic for future work. Other: L40: No need to capitalize “entropic” L46, “privacy constraints:” Is it clear the technique in this paper has any privacy-preserving properties? L150: [this comment is likely a mistake on my part but I’ll ask it anyway!] I’m not sure I follow why sqrt(W)*p=0 implies that all the p’s are equal. For example, consider the line graph with vertex labels 0-1-2-3-4-5-6. Notice the label of every vertex on the graph is the average of the labels of its neighbors, but the signal is not a constant. Why does that not happen for this paper? To fix the issue on the previous line, I would think a simpler way to write the constraint is Dp=0, where D is the E x V incidence matrix that takes one value per vertex and subtracts along edges; one way to obtain the graph Laplacian is by writing D^T * D = L. I believe the rest of the algorithm would work out identically. It seems a large part of sec 3.1-3.2 is a standard application of proximal methods (problem (P) is a standard one of course); perhaps it could be shortened if the authors need to expand their discussion elsewhere. In particular, two different algorithms are given which are both at a level of generality greater than what is needed later in the paper. Currently this section and sec 4 are quite dense and hard to follow relative to the other discussion. Given that the optimization problem is entropically regularized, why in sec 3.2 is it chosen to use quadratic terms for the proximal algorithm? I would have guessed entropy/KL would be a better fit here. Figure 1 is a great idea! It answer’s my main question, which is dependence of convergence on topology of the graph. What caught me by surprise is that convergence for the star graph is so slow --- I would have expected faster “mixing time” (I know, not in the MCMC sense!) since all the distributions are two links away in the topology. Is there intuition for why this doesn’t happen? [took a look at the rebuttal and it's all reasonable to me --- this is a nice paper!]

Reviewer 2



SUMMARY The authors introduce a new algorithm to compute in a distributed manner the (regularized) Wasserstein barycentre (on a fixed grid) of a family of distributions that can be sampled from. The authors reformulate elegantly the problem using the Fenchel conjugate of the entropy-regularized Wasserstein loss. They solve this problem with an accelerated stochastic gradient method on the dual and show how to recover the barycentre from these iterations, with precise error bounds. They conclude with experimental results realized on various distributed network configurations and on various images datasets. QUALITY Compared to previous methods, the approach and analysis adopted in this article allows to greatly reduce the computations and communication needed once a new sample is observed. This leads to an efficient way to compute Wasserstein barycenters, even in the undistributed setting or in the discrete measures setting. The experimental results are relevant and agree with the theoretical part. CLARITY The related work and contribution section is particularly well written. I have doubts about the relevance of introducing an abstract setting in Section 3, but this might be a matter of taste. Also, I have a tendency to prefer the "unaccelerated variant" of an algorithm to be introduced first because the accelerated steps makes the pseudo-code much harder to parse. ORIGINALITY The ideas presented here a quite natural from a distributed computing point of view, but are assembled for the first time for the computation of Wasserstein barycenter. The dual formulation is elegant and the authors had to prove new results to convert the dual convergence guarantee into primal guarantees. SIGNIFICANCE In my opinion, this paper constitutes a reference method for the distributed computation of barycenters. The main limitation of the setting is the need for a fixed discrete support for the barycenter, but removing this constraint is certainly challenging. SPECIFIC REMARKS Here I list some remarks or suggestions: - l.21 : I find the explanation a bit strange and, as such, brings opacity (this statement could apply to any "length distance"). - I would state somewhere in the introduction that support of the unknown barycenter is fixed in advance. - l.134 : I am not sure that this changes anything (I guess that the solution does not change as long as the reference measure has a density with respect to the uniform distribution that is of the form f(y)*g(z) ) - l.153 : maybe say that this is the Kronecker product - l.203 : replace Phi(x,xi) by Phi(lambda,xi). Although I understand that it stands for "equivalence of oracle", the equality sign here is somewhat misleading. - l.207 : gradient - Theorem 1 : norm should be the norm in E. The output eta_N does not appear in (6), so it is void to say that it "satisfies" (6). - in Algorithm 1, the primal iterate x can be computed offline (given the sequences lambda, xi and tau ) : if I understand well, this algorithm is simply accelerated stochastic gradient method on the dual problem, but you are able to transfer the error bounds to the primal variable/objective using primal-dual relationships. If I am correct, then I think it would be much more clear if presented this way. In particular, I find the name "primal-dual" slightly misleading. - l.293 : missing space - I would have liked a comparison with unaccelerated stochastic gradient [UPDATE AFTER REBUTTAL I am still confident that this paper is a good contribution to NIPS and I am very satisfied with authors' response. In particular, the theoretical issue raised by reviewer #3 seems to be resolved.]

Reviewer 3



This paper aims at solving the regularised Wasserstein barycentre problem for the semi-discrete case. It assumes the distribution of the probability measures over a network and formulate the problem as the minimisation of a convex objective under equality constraints. The equality constraint is formulated via the Laplacian of the graph representing the network, so that information flow is only allowed between agents that are connected with each other. To solve this problem, the authors develop a general primal dual method for stochastic optimization and apply it to the original distributed barycentre problem. Algorithm 1 is essentially the accelerated method applied to the dual problem with stochastic gradient. The obtained theoretical result is quite surprising as usually solving the dual problem will not lead to a bound as good as (6). If we look at the proof, we found that equation (34) should not be derived from (33). It is then not clear how the authors get (34) and then (37). Unless this point is clarified, the results claimed in Theorem 1 are not proved correctly and the theoretical part of the paper does not hold. ---This part has been clarified in authors' feedback. Minor comments: -typos in eqn. 30, 31, 33.